# Comparative analysis of the syncytiotrophoblast in placenta tissue and trophoblast organoids using snRNA sequencing

Madeline M Keenen[1], Liheng Yang[2], Huan Liang[1,3], Veronica J Farmer[1], Rizban E Worota[2], Rohit Singh[1,3], Amy S Gladfelter[1]*, Carolyn B Coyne[2,4]*

[1]Duke University School of Medicine, Department of Cell Biology, Durham, United States; [2]Duke University School of Medicine, Department of Integrative Immunobiology, Durham, United States; [3]Duke University School of Medicine, Department of Biostatistics and Bioinformatics, Durham, United States; [4]Duke Human Vaccine Institute, Durham, United States

*For correspondence:
amy.gladfelter@duke.edu (ASG);
carolyn.coyne@duke.edu (CBC)

Competing interest: The authors declare that no competing interests exist.

## eLife Assessment

This **important** study uses single-cell transcriptomics to analyze syncytiotrophoblasts in two trophoblast organoid models compared to primary placental tissue, providing **compelling** insights into syncytialization and highlighting the utility of organoid models in placental research. It also serves as an invaluable resource for the field.

**Abstract** The syncytiotrophoblast (STB) is a multinucleated cell layer that forms the outer surface of human chorionic villi. Its unusual structure, with billions of nuclei in a single cell, makes it difficult to resolve using conventional single-cell methods. To better understand STB differentiation, we performed single-nucleus and single-cell RNA sequencing on placental tissue and trophoblast organoids (TOs). Single-nucleus RNA-seq was essential for capturing STB populations, revealing three nuclear subtypes: a juvenile subtype co-expressing CTB and STB markers, one enriched in oxygen sensing genes, and another in transport and GTPase signaling. Organoids grown in suspension culture (STBout) showed higher expression of STB markers, hormones, and a greater proportion of the transport-associated nuclear subtype while TOs grown with an inverted polarity (STBin) exhibited a higher proportion of the oxygen sensing nuclear subtype. Gene regulatory analysis identified conserved STB markers, including the chromatin remodeler RYBP. Although RYBP knockout did not impair fusion, it downregulated CSH1 and upregulated oxygen-sensing genes. Comparing STB expression in first trimester, term, and TOs revealed shared features but context-dependent variability. These findings establish TOs as a robust platform to model STB differentiation and nuclear heterogeneity, providing insight into the regulatory networks that shape placental development and function.

## Introduction

During the course of human gestation, the developing fetus forms an entire external organ to support its growth—the placenta. While the fetal organs undergo development, the placenta assumes a multifaceted role, serving to facilitate molecular exchange, perform essential metabolic functions, produce

hormones, prevent loss of immune tolerance, and act as a barrier against the vertical transmission of pathogens (*Aye et al., 2022*; *Benirschke et al., 2012*; *Costa, 2016*; *Megli and Coyne, 2022*). The placenta's remarkable functional complexity is underscored by its distinctive cellular architecture. Its outer layer encompasses a giant single cell called the syncytiotrophoblast (STB), that contains billions of nuclei and envelops the chorionic villi (*Barker et al., 1973*; *Burton and Jauniaux, 1995*; *Haeussner et al., 2014*). The STB is formed via cell-cell fusion of the underlying cytotrophoblast (CTB) cell population. CTBs reside on the basement membrane of chorionic villi to contribute new nuclei into the STB or lie at the interface between the placenta villi and the maternal decidua in multi-cell layered structures termed cell columns (CTB-CC; *Figure 1A*; *Boyd and Hamilton, 1970*). CTBs closer to the maternal decidua are more differentiated than their counterparts lower in the column and eventually undergo epithelial-to-mesenchymal transition (EMT) to form fully differentiated extravillous trophoblast (EVT) cells that invade into the decidua (*Figure 1A*; *Arutyunyan et al., 2023*; *Turco and Moffett, 2019*). The molecular mechanisms and environmental cues that drive CTB differentiation into either STB or EVT lineages is an area of active research and defining these trajectories are essential to understand placenta development and pathogenesis.

Several groups have applied single-cell RNA sequencing (SC) of primary tissue throughout gestation, trophoblast organoids (TOs), and trophoblast stem cells (TSCs) to characterize EVT differentiation (*Arutyunyan et al., 2023*; *Li et al., 2024*; *Liu et al., 2018*; *Marsh et al., 2022*; *Pique-Regi et al., 2019*; *Shannon et al., 2024*; *Suryawanshi et al., 2018*; *Vento-Tormo et al., 2018*). This has generated a lineage map of CTB-to-EVT differentiation with identification of its terminal states and the potential transcription factors (TFs) involved. However, these datasets contain very few cells from the STB, limiting characterization of the CTB-to-STB differentiation process. This scarcity likely arises because the STB, being a large single cell, is excluded during the single-cell isolation and size filtration steps required for the 10 x Genomics microfluidics pipeline. Therefore, approaches like single-nucleus RNA sequencing (SN) may be essential to properly capture the heterogeneity of STB gene expression. In fact, a recent study performed SN on placenta tissue (six first trimester and six full-term tissues) and captured STB nuclei and defined their lineage trajectories at each gestational age (*Wang et al., 2024*). Their analysis suggests that the STB can bifurcate into at least two nuclear lineages post-fusion, associated with either hormone expression and GTPase signaling or with an oxygen response. This implies different functions of the STB may be attributed to distinct individual nuclei within the same giant cell. However, how nuclei with distinct gene expression arise and how they impact the function of the entire STB cell is not known. Dissecting nuclear heterogeneity in the STB will require a molecular biology and genetic toolkit that has been largely inaccessible for human pregnancy models.

A major challenge in establishing genetically tractable and accessible models for the human placenta is the remarkable diversity of placental structures amongst mammals, and notably the variations in the tissue architecture and cell types seen even between humans and mice (*Hemberger et al., 2020*; *Wooding and Burton, 2008*). In the last several years, TOs have emerged as powerful tools for studying trophoblast differentiation. CTB progenitor cells can be isolated from tissue throughout gestation, as CTB remain mitotic throughout pregnancy (*Haider et al., 2018*; *Mayhew, 2014*; *Turco and Moffett, 2019*; *Yang et al., 2022*). They can subsequently be maintained in a proliferative state capable of trophoblast differentiation by using a growth factor cocktail and cultivation in extracellular matrix (*Okae et al., 2018*; *Turco et al., 2018*; *Yang et al., 2022*), and spontaneously fuse to form STB (*Haider et al., 2018*; *Li et al., 2023*; *Turco and Moffett, 2019*; *Yang et al., 2022*). In standard culture conditions, TOs exhibit an inverted architecture compared to placental villi in vivo, with an outward facing proliferative CTB layer and a largely inward facing STB (STBin; *Haider et al., 2018*; *Turco and Moffett, 2019*; *Yang et al., 2022*). Our recent work has established a method to reverse the cellular polarity of TOs to their native orientation, resulting in organoids containing very large (>50 nuclei) STB on the outermost layer and mononuclear CTBs positioned in the center (STBout), and exhibit increased secretion of the STB-associated hormone human chorionic gonadotropin (hCG) (*Yang et al., 2024*). Recapitulating the native orientation of outward facing STB is essential to model key aspects of STB function in vitro, such as molecular transport and its role as a barrier to infection and immune cells. However, how changes in TO orientation affect the functional differentiation of STB remains an important and unexplored area of investigation. While the majority of CTBs in TOs differentiate into the STB, a small proportion can spontaneously differentiate into HLA-G+ EVTs. This EVT percentage can be increased through a three-step treatment involving Neuregulin-1 (NRG1) (EVTenrich;

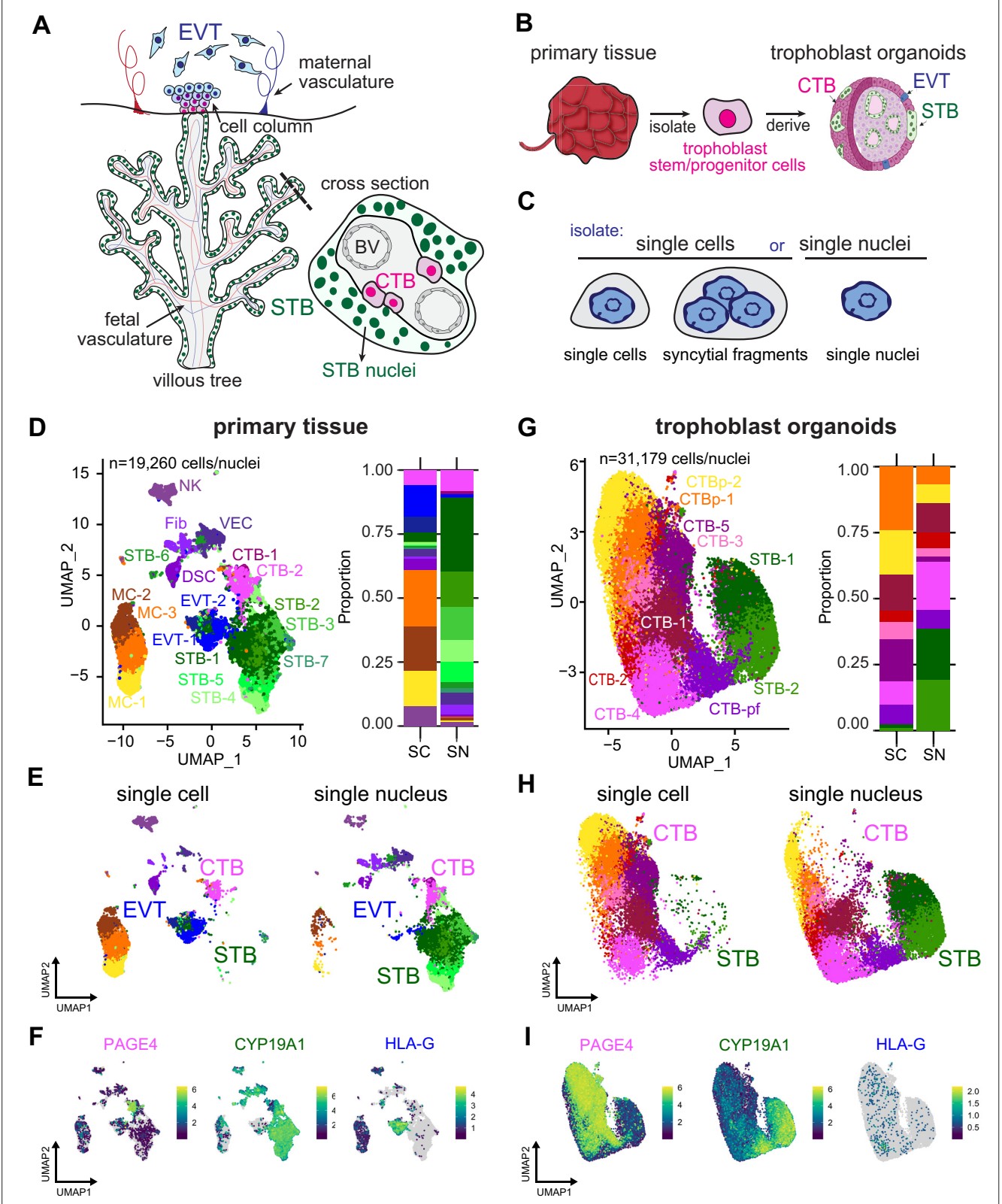

**Figure 1.** Single-nucleus sequencing is essential to capture the STB lineage in full-term tissue and TOs. (**A**) Cartoon of placenta villous tree and cross-section, with the multinucleated STB highlighted in green, progenitor CTB cells in magenta, and extravillous cytotrophoblasts in blue. (**B**) Schematic of organoid generation via isolation of trophoblast progenitor cells from full-term placental tissue. (**C**) Schematic of experimental setup. Both tissue and TOs were processed into either single cells/syncytial fragments or single nuclei and sequenced. UMAP of integrated SC and SN datasets

*Figure 1 continued on next page*

*Figure 1 continued*

collected from primary tissue (**D**) and TOs (**G**). Cell/nucleus types are annotated as follows: cytotrophoblast (CTB), cytotrophoblast pre-fusion (CTB-pf), syncytiotrophoblast (STB), extravillous trophoblast (EVT), dendritic stem cell (DSC), vascular endothelial cell (VEC), fibroblasts (Fib), natural killer cell (NK), and macrophage (MC). Cell/nucleus types that contained multiple clusters are identified with -number after the name. (**E and H**) UMAP of the integrated SC and SN dataset separated into individual UMAPs by single cell or nucleus processing type for primary tissue (**E**) and TOs (**H**). (**F and I**) Expression of key trophoblast markers, including PAGE4 (CTB), CYP19A1 (STB), and HLA-G (EVT) in primary tissue (**F**) and TOs (**I**).

The online version of this article includes the following figure supplement(s) for figure 1:

**Figure supplement 1.** Characterization of cell/nucleus types in the integrated SC/SN dataset.

**Figure supplement 2.** Characterization of cell types in the full-term tissue SC dataset.

**Figure supplement 3.** Characterization of nucleus types in tissue SN dataset.

**Figure supplement 4.** UMAP of the integrated SC and SN dataset separated into individual UMAPs by single cell or nucleus processing type for primary tissue.

*Haider et al., 2018*; *Turco and Moffett, 2019*; *Yang et al., 2022*). Consequently, TOs offer the ability to induce differentiation along both STB and EVT lineages and have rapidly become a powerful and accessible tool to investigate trophoblast biology.

As TOs become increasingly prevalent as a research model, it is crucial to assess their resemblance to in vivo trophoblast cell types. Further, the gene expression signature of the STB and any heterogeneity that exists amongst nuclei remains enigmatic due to its unique multinucleated ultrastructure, both in vitro and in vivo. In this study, we performed comparative SC and SN on primary full-term placenta tissue and TOs. We found that SN is essential to capture STB gene expression both in tissue and TOs, while SC enriches for mitotic cells, maternal immune cells, and EVT cells in tissue. Differential gene expression and pseudotime analysis of distinct STB nuclei in TOs identified three distinct subtypes reminiscent of those recently identified in vivo: a juvenile population that exhibits both CTB and STB expression, an FLT1-expressing population enriched in genes involved in oxygen sensing, and a subtype enriched in expression of molecular transport and GTPase signaling molecules. While STB$^{in}$ and STB$^{out}$ conditions maintain a similar proportion of CTB cell nuclei, STB$^{out}$ TOs exhibited a higher proportion of the transport and GTPase STB-3 nuclear subtype while STB$^{in}$ exhibited a higher proportion of the oxygen sensing STB-2 subtype. Pseudotime and gene regulatory network analysis of RNA velocity identified genes linked to STB differentiation, including the chromatin effector RYBP which is enriched in STB$^{out}$ TOs. To validate this analysis, we utilized CRISPR/Cas9 to knock-out RYBP in TOs. We found that deletion of RYBP in STB$^{in}$ TOs did not affect cell-cell fusion or STB formation, but bulk RNA sequencing demonstrated that RYBP KO resulted in a significant decrease in the expression of the pregnancy hormone CSH1 and an increase in expression of key genes that define the oxygen sensing STB-2 nuclear subtype. Finally, STB gene expression was compared between TOs and primary tissue at different stages of gestation. The CTBs were remarkably similar across all conditions, indicating that CTBs isolated from term placenta that are used for TO generation are comparable to those isolated at earlier stages of pregnancy. The STB displayed both commonalities and notable variability across the sample types. Together, this work demonstrates the capacity of TOs to mirror STB differentiation and the nuclear subtypes seen in vivo, providing an accessible platform to dissect the key molecular pathways underlying placenta function, distinct STB subtypes, and trophoblast-related pregnancy disorders.

## Results

### Single-nucleus sequencing is necessary to capture the gene expression signature of STBs in placental tissue and TOs

In this study, we set out to compare the transcriptional profile of full-term placental tissue composed of placental villi and decidua (*Figure 1A and B*) to STB$^{in}$ TOs previously derived from placental tissue (*Figure 1B*). To assess whether SC or SN sequencing could effectively capture trophoblast cell populations, we processed matched samples into single cells/syncytial fragments or single nuclei (*Figure 1C*, Materials and methods). The SC and SN datasets generated from primary placental tissue (*Figure 1D–F*) and TOs (*Figure 1G–I*) were integrated, and a graph-based clustering approach used to identify clusters (*Butler et al., 2018*; *Hao et al., 2021*; *Satija et al., 2015*; *Stuart et al., 2019*). Each

dataset was visualized with a UMAP plot (*Figure 1D and G*) and established gene expression markers were used to determine the cell or nuclear identity of each cluster (*Figure 1F and I*, *Figure 1—figure supplement 1C and F*, and *Supplementary files 1-2*; *Arutyunyan et al., 2023*; *Derisoud et al., 2024*; *Vento-Tormo et al., 2018*).

To determine how each sequencing technique affects the detection of trophoblast cell types, we first defined the cell/nucleus types in the primary tissue dataset. We identified two CTB clusters, seven STB clusters, and two EVT clusters, each expressing their respective markers (e.g. PAGE4, CYP19A1, and HLA-G; *Figure 1D and F*, *Figure 1—figure supplement 1C*). The clusters for each cell type are labeled with a randomly ordered number. There was a similar distribution of these subtypes across all three donor tissues (*Figure 1—figure supplement 1A–B*). As in previous single-cell approaches, our SC tissue dataset captured only a small fraction of the STB cell type (6% of total cells are STB). In contrast, the predominant population in the SN preparation was STB nuclei, accounting for 76% of the total nuclei count, (*Figure 1D-E*, *Figure 1—figure supplement 1A, B*) and consistent with their relative prevalence in vivo (*Mayhew, 2014*; *Mayhew and Simpson, 1994*; *Simpson et al., 1992*). Specifically, the SN dataset on primary tissue had an 8:1 ratio of STB to CTB, resembling stereological estimates of placental trophoblast compositions at full-term (9:1) indicating that enzymatic digestion of nuclei is capturing a representative population of these trophoblast nucleus types. (*Figure 1D*). While the STB subtype is better captured by SN sequencing, SC sequencing exhibited significant enrichment in both EVT (18% of total in SC vs. 1.3% in SN) and macrophages (53% of total in SC vs. 1.9% in SN) (*Figure 1D–E* and *Figure 1—figure supplement 1A–B*). As a control for SC/SN dataset integration, cell/nucleus types were identified within the individual SC and SN datasets and proportions mirrored the integrated SC/SN dataset (*Figure 1—figure supplements 2 and 3*). Thus, differences between tissue processing for either SC or SN impact the cell types that can be recovered and subsequently sequenced and SN is essential for the analysis of the STB.

We next defined the nucleus types represented in the STB$^{in}$ TO dataset, in which we identified two proliferating CTB clusters (CTB-p), five non-proliferative CTB clusters (CTB-1–5), one pre-fusion CTB cluster with high expression of endogenous retroviral fusion genes (CTB-pf), and two STB clusters (STB 1–2; *Figure 1G-I*, *Figure 1—figure supplement 1F*). The clusters for each cell type are labeled with a randomly assigned number and color, and therefore cluster numbers and color for each cell type cannot be directly compared between the tissue and TO individual datasets until they are integrated in Figure 6. Like placenta tissue, the SC dataset captured only a small number of the total STB present in TOs, with only 2.4% of the total cell population attributed to the STB (*Figure 1G*, *Figure 1—figure supplement 1E*). In contrast, the STB accounted for 38% of the total nuclear numbers captured by SN sequencing (*Figure 1G*, *Figure 1—figure supplement 1E*). The differences in non-proliferative CTB populations (CTB 1–5) between SC and SN sequencing were less pronounced, with 57% of the total cell population from SC sequencing attributed to these cells and 56% of nuclei in the SN dataset (*Figure 1G*, *Figure 1—figure supplement 1E*). Similarly, both SC and SN captured nearly identical numbers of CTB-pf cells (7%). However, the number of CTB-p were only 16% of the total population in SN despite accounting for 40% in SC, consistent with the challenge to isolate nuclei from mitotic cells for SN sequencing due to the breakdown of the nuclear envelope during mitosis (*Figure 1G*). Collectively, this comparison underscores the necessity of SN sequencing to representatively capture the gene expression of the STB population in both primary placental tissue and TOs. However, other cell types including mitotic CTB populations (TOs), EVT (tissue), and macrophages (tissue) are enriched in SC and demonstrate a combined SC/SN approach is necessary to capture every cell type in the human placenta or TOs.

## Defining trophoblast lineage composition in distinct TO culture conditions

The STB has been historically understudied due to the challenges of its multinucleate architecture. Therefore, we next investigated the impact of TO culture conditions on trophoblast lineage composition using only SN sequencing because it more accurately captured the STB population. To do this, we used TOs isolated from three unique placentas and cultured them each in three distinct culture conditions—standard Matrigel conditions to generate STB$^{in}$ TOs, in suspension to generate STB$^{out}$ TOs, and with NRG1 to enrich for EVT cells (EVT$^{enrich}$; *Figure 2A–C*, Materials and methods, *Supplementary files 3-4*). Each organoid condition was processed into suspensions of nuclei, SN sequenced

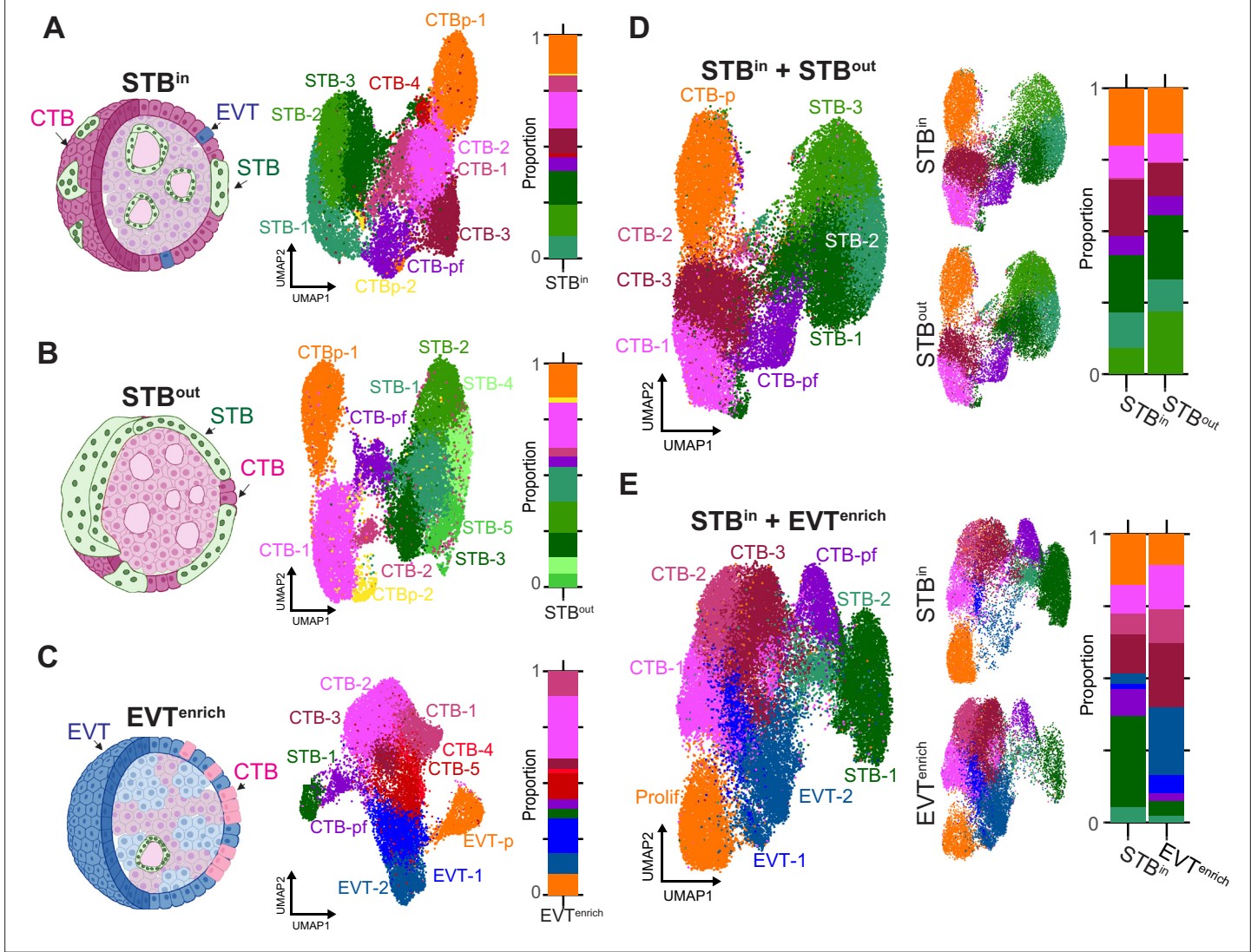

**Figure 2.** Comparison of TO gene expression in different culture conditions by SN RNA sequencing. TOs were grown with the STB facing inward (STB$^{in}$), outward (STB$^{out}$), or induced to differentiate into EVT (EVT$^{enrich}$). Schematic of TO nucleus type composition (left), UMAP of the individual SN datasets (middle), and barplot showing the proportions of each nucleus type (right) are shown for STB$^{in}$ (**A**), STB$^{out}$ (**B**), or EVT$^{enrich}$ (**C**). UMAP of the integrated datasets (left), integrated dataset split by culture condition (middle), and barplot showing the proportions of each nucleus type (right) are shown for the STB$^{in}$ +STB$^{out}$ integrated dataset (**D**) or STB$^{in}$ +EVT$^{enrich}$ dataset (**E**). Prolif in E refers to proliferative subtype encapsulating both EVT-p and CTB-p.

The online version of this article includes the following figure supplement(s) for figure 2:

**Figure supplement 1.** Characterization of nucleus subtypes in individual TO datasets.

**Figure supplement 2.** Characterization of nucleus types in the merged TO SN datasets.

**Figure supplement 3.** Comparison of TO and TSC SN datasets.

in parallel, and data from each biological replicate integrated, clustered, and plotted on a UMAP. (*Figure 2A–C*, *Figure 2—figure supplement 1A*, *Supplementary files 1-2*). The clusters for each cell type are labeled with numbers assigned in a random order.

We first assessed how CTB subtypes varied amongst the culturing methods that generated different organoid organizations. We found five populations of CTBs that were identified using well-established markers, such as CDH1 and TENM3, and accounted for 53% of the total population in STB$^{in}$, 38% in STB$^{out}$, and 57% in EVT$^{enrich}$ (*Figure 2A–C*, *Figure 2—figure supplement 1B*). These CTBs could be delineated into multiple subtypes expressing proliferative markers (KI67 and PCNA in CTB-p), CTB cell column (CTB-CC) markers (LPCAT1, NOTCH1, and ITGA2), or pre-fusion intermediate markers (retroviral fusion protein ERVFRD-1 and GREM2 in CTB-pf; *Figure 2—figure supplement 1B*). STB$^{in}$

and STB<sup>out</sup> TOs both contained two proliferative CTB populations (CTBp-1 and CTBp-2) that together accounted for 18% of the total nuclear population of each dataset (*Figure 2A–B*). In contrast, there was a single proliferative population in EVT<sup>enrich</sup> that accounted for 10% of the population, was closer to EVT than CTB on the UMAP, and had downregulated CTB markers and upregulated EVT markers including HLA-G and MMP2. This suggests that TO-derived differentiated EVTs undergo mitosis, as observed in vivo (*Figure 2C*, *Figure 2—figure supplement 1B, D*; *Arutyunyan et al., 2023*). In the STB<sup>in</sup> condition, CTB-2 expressed canonical CTB-CC markers and accounted for 17% of the total nuclear population (*Figure 2—figure supplement 1B–C*). The STB<sup>out</sup> condition predominantly consisted of a single non-proliferative CTB population (CTB-1). A subset of this cluster expressed the CTB-CC marker ITGA2 suggesting this CC population was still present but not identified as a separate cluster (*Figure 2—figure supplement 1B–C*). In contrast, three of the five CTB populations in EVT<sup>enrich</sup> TOs expressed the CTB-CC markers ITGB6 and LPCAT1 (*Figure 2—figure supplement 1B–C*), consistent with the function of CTB-CCs to differentiate into EVT (*Turco and Moffett, 2019*). Finally, CTB-pf accounted for 4% of the total population in STB<sup>out</sup>/EVT<sup>enrich</sup> TOs and 6% of STB<sup>in</sup>, suggesting culture conditions did not dramatically change the proportion of this intermediate cell type (*Figure 2A–C*, *Figure 2—figure supplement 1B*). Thus, while most CTB identities are present across conditions, there are notable differences in the proportion of different CTB subtypes depending on culture conditions.

We next sought to determine how culture conditions influenced the differentiation of CTBs into either the STB or EVTs. In each culture condition STB nuclei were present, accounting for 39% (STB<sup>in</sup>), 53% (STB<sup>out</sup>), and 4% (EVT<sup>enrich</sup>) of the total population (*Figure 2A–C*). As anticipated, few to no EVTs were present in STB<sup>in</sup> and STB<sup>out</sup> TOs (*Figure 2A–B*; *Turco and Moffett, 2019*; *Yang et al., 2024*; *Yang et al., 2022*). In contrast, EVTs accounted for 24% of the total population in EVT<sup>enrich</sup> TOs and separated into two clusters (EVT-1, EVT-2) (*Figure 2C*). Each had an increased expression of the mature EVT markers HLA-G, MMP2, and DIO2 (*Figure 2—figure supplement 1B*). Together, this demonstrates that STB<sup>out</sup> culture conditions promotes further STB differentiation while EVT<sup>enrich</sup> conditions promotes EVT differentiation.

To directly compare the relative populations and gene expression between TO culture conditions, we integrated the STB<sup>in</sup> dataset with either STB<sup>out</sup> or EVT<sup>enrich</sup> TOs (*Figure 2D–E*). Each nucleus type retained the expression of canonical markers (*Figure 2—figure supplement 2A–B*), but EVT<sup>enrich</sup> exhibited higher basal expression of the EVT markers HLA-G and DIO2 (*Figure 2—figure supplement 2C*). The nucleus types in the STB<sup>in</sup> +STB<sup>out</sup> dataset showed significant overlap on the UMAP, indicating relatively consistent gene expression across culture conditions (*Figure 2D*). However, the proportions were not identical for each nucleus type and reflected the differences observed in the individual datasets. Of note, there was a higher percentage of CTB-3 that expressed CTB-CC markers in STB<sup>in</sup> (20% of STB<sup>in</sup> and 11% of STB<sup>out</sup>) and a decrease in the STB-3 population (9% of STB<sup>in</sup> and 20% of STB<sup>out</sup>; *Figure 2D*, *Figure 2—figure supplement 2A*) suggesting that STB<sup>out</sup> culture conditions promotes STB differentiation and prevents CTB-CC differentiation down the EVT lineage. Despite CTB-pf accounting for 7% of each dataset in STB<sup>in</sup> and STB<sup>out</sup>, the STB:CTB ratio was nearly halved in STB<sup>in</sup> compared to STB<sup>out</sup> (1.3:1 in STB<sup>in</sup> versus 2.5:1 in STB<sup>out</sup>). This indicates a higher proportion of nuclei in STB<sup>out</sup> TOs have undergone cell-cell fusion to become STB. Similarly, STB<sup>out</sup> TOs expressed higher levels of key STB markers and hormones, indicating culture in suspension promotes enhanced STB differentiation (*Figure 2—figure supplement 2D* and *Figure 3—figure supplement 1D*). These results are consistent with our previous report demonstrating an increase in the number of STB nuclei and enhanced expression of hCG- genes in the STB<sup>out</sup> condition (*Yang et al., 2024*). In contrast, the STB<sup>in</sup> +EVT<sup>enrich</sup> merged dataset exhibited significantly less overlap on the UMAP and amongst nucleus type proportions (*Figure 2E*). In particular, STB<sup>in</sup> TOs had a dramatic increase in STB nucleus types (37% of STB<sup>in</sup> and 7% of EVT<sup>enrich</sup>) while EVT<sup>enrich</sup> exhibited an increase in EVT types (5% of STB<sup>in</sup> and 30% of EVT<sup>enrich</sup>; *Figure 2E*, *Figure 2—figure supplement 2B*), validating EVT<sup>enrich</sup> conditions promote CTB differentiation into EVT instead of STB.

Lastly, we compared STB<sup>out</sup> TOs with a publicly available SN dataset from trophoblast stem cell (TSC) derived STB to identify differences in STB populations that might exist between these models (*Wang et al., 2024*). We integrated the STB<sup>out</sup> TO dataset with the TSC dataset, performed clustering, and visualized the results on a UMAP. Both TO and TSC models showed a nearly equivalent composition of trophoblast nucleus types, including CTB-p, CTB-pf, EVTs, and two distinct STB populations (STB-1 and STB-2; *Figure 2—figure supplement 3A–C*). Despite similarities in nuclear

proportions, pseudobulk differential expression analyses revealed significant differences in the transcriptional profiles of CTB-pf and STB populations derived from TOs and TSCs. TSC-derived CTB-pf and STB clusters highly expressed the EVT marker HLA-G, which was absent in TO-derived CTB-pf and STBs (*Figure 2—figure supplement 3D*). Additionally, TO-derived CTB-pf and STBs showed significantly higher expression of STB-associated hormones and other factors, including PSGs, CGBs, CSH1, HOPX, and KISS1 (*Figure 2—figure supplement 3D–E*; *Costa, 2016*). Therefore, while the two models exhibit a similar proportion of nucleus types, there are notable gene expression differences within each.

Together, these data highlight that culture conditions not only influence the composition of trophoblast nucleus types but also drive their differentiation into distinct lineages, consistent with previous reports and underscoring the critical role of the environmental cues present in each culture condition in shaping trophoblast identity (*Arutyunyan et al., 2023*; *Turco and Moffett, 2019*; *Yang et al., 2024*; *Yang et al., 2022*).

## Comparative transcriptional profiling reveals three distinct STB subtypes with varied proportions in STB$^{in}$ and STB$^{out}$ TOs

Given the increased proportion of STB nuclei in STB$^{out}$ TOs and enhanced expression of typical STB markers in STB$^{out}$ TOs, we next sought to identify the genes that define each STB subtype. To do this, we utilized the merged STB$^{in}$ +STB$^{out}$ dataset that contained three STB subpopulations. This dataset exhibited differential enrichment of each STB subpopulation within the two culture conditions (*Figure 3A*). We first analyzed the genes enriched in each STB subtype and identified hundreds of genes whose expression was conserved in the STB subtypes of both STB$^{in}$ and STB$^{out}$ TOs (*Figure 3B*, *Figure 3—figure supplement 1A*, *Supplementary file 8*). STB-1 is closest to CTB-pf on the UMAP and expressed many genes previously associated with CTBs, including the TFs TEAD1 and TP63 (*Figures 2D and 3B*; *Li et al., 2014*; *Mizutani et al., 2022*). In fact, when the top genes in STB-1 were plotted as a dotplot for every nucleus type in the dataset these genes were most enriched in the CTB subtypes (*Figure 3—figure supplement 1B*). The gene ontology (GO) terms associated with STB-1-enriched genes are involved in RNA splicing, stem cell maintenance, and RAS signaling (*Figure 3C*, *Figure 3—figure supplement 1C*). Candidate genes from each of these GO terms demonstrated expression predominantly in CTBs, intermediate expression in STB-1, and lower expression in the remaining two STB populations (*Figure 3—figure supplement 1C*). STB-2 expressed a unique subset of genes that included the VEGF receptor FLT1, the TGFβ family member INHBA, and the insulin regulator PAPPA2, associating this subtype with ER stress and oxygen sensing (*Figure 3B–C*; *Barrios et al., 2021*; *Li et al., 2022*; *Sasagawa et al., 2021*; *Sasagawa et al., 2018*). Female pregnancy was identified as an enriched GO term in STB-2 (*Figure 3C–D*) and many of the genes found in this term are linked to angiogenesis. This included eight pregnancy specific glycoprotein (PSG) paralogs that are pro-angiogenic, VEGFA, and the growth factor Angiopoietin-2 that facilitates vascular development in specific contexts (*Akwii et al., 2019*, *Moore et al., 2021*; *Supplementary file 9*). Interestingly, STB-2 was more abundant in STB$^{in}$ than STB$^{out}$ (*Figure 3A*). We validated this finding with RNA fluorescence in-situ hybridization (FISH) and found an increase in the expression of the STB-2 marker PAPPA2 in STB$^{in}$ TOs compared to STB$^{out}$ (*Figure 3D and F* and *Figure 3—figure supplement 3A*). Finally, the top genes in STB-3 included the sodium/bicarbonate transporter SLC4A4, the matrix metalloprotease ADAMTS6, the collagen receptor ITGA1, and protein kinase C epsilon PRKCE (*Figure 3B*; *Cain et al., 2022*; *Cain et al., 2016*; *Zeltz and Gullberg, 2016*). GO terms demonstrated that the STB-3 cluster was enriched for processes involved in GTPase signaling, vascular transport, and actin organization (*Figure 3C*, *Figure 3—figure supplement 1C*). Both the vascular transport and transport across the blood brain barrier GO terms were enriched for molecules involved in the transport of small molecules, amino acids, and fatty acids (*Supplementary file 9*). In contrast to STB-1 and STB-2, the percentage of STB-3 nuclei were doubled from 20% in STB$^{in}$ to 40% in STB$^{out}$. We validated this finding via RNA-FISH using the STB-3 marker ADAMTS6 and observed increased RNA expression in STB$^{out}$ compared to STB$^{in}$ TOs (*Figure 3E and G* and *Figure 3—figure supplement 3B*). Interestingly, most pregnancy hormones were expressed at similar levels in the STB-2 and STB-3 subtypes but exhibited increased expression in STB$^{out}$ compared to STB$^{in}$ (*Figure 3—figure supplement 2C–D*). In summary, STB-1 exhibited intermediate gene expression between CTB and STB, STB-2 expressed vascular signaling/oxygen sensing factors, while STB-3 was enriched in transport/GTPase signaling

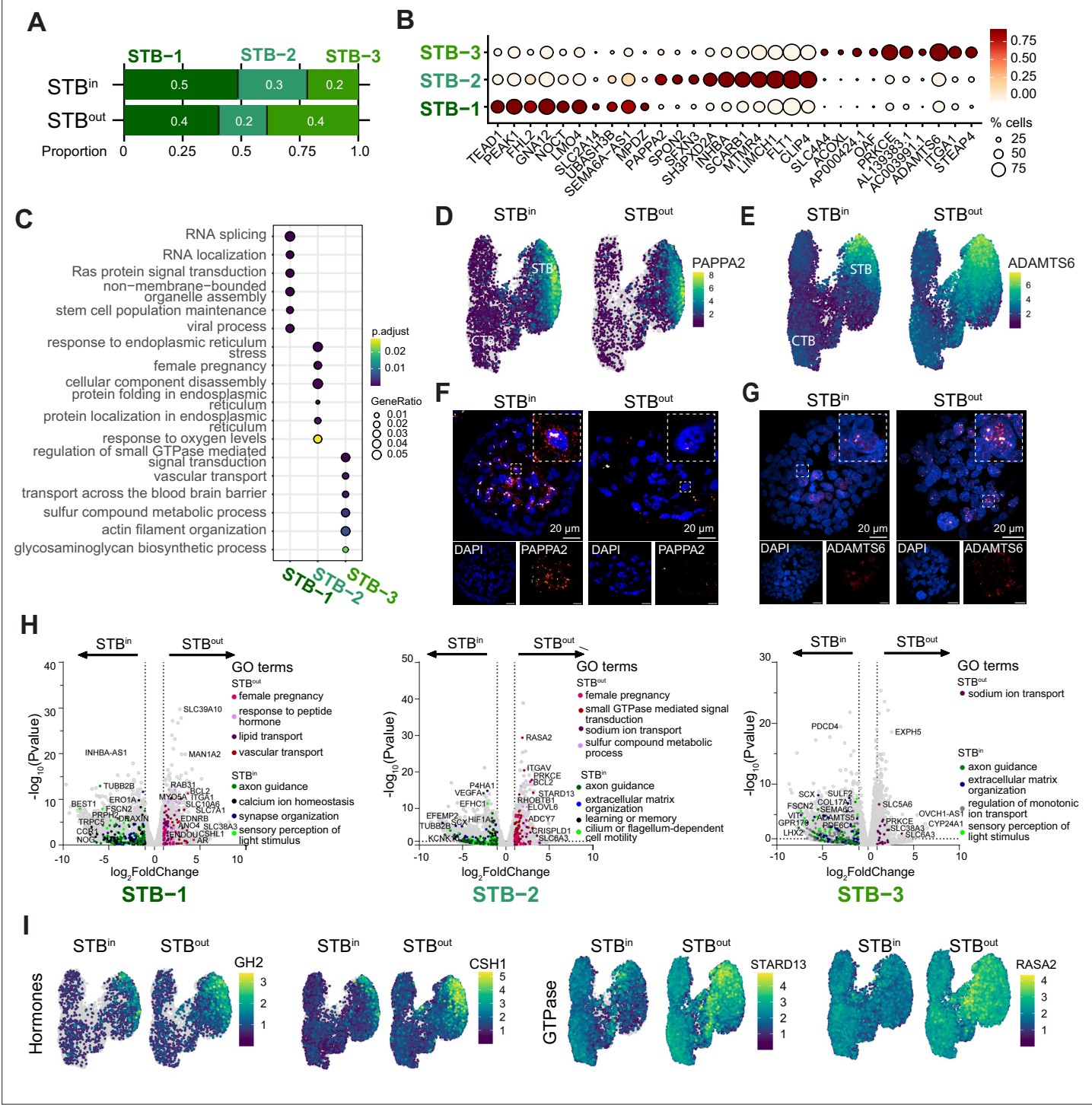

**Figure 3.** STB subtype analysis and expression differences in STBin vs STBout TOs. (**A**) Proportion of each STB subtype in the integrated STBin +STBout dataset. (**B**) Top differentially expressed genes (DEGs) of each STB subtype is shown as a dot plot. Mean expression is shown as a color scale and percent of nuclei expressing each gene is demonstrated by size of the dot. (**C**) Top Biological Process GO terms associated with DEGs in each STB subtype was determined with clusterProfiler. (**D**) Feature Plots of PAPPA2 in STBin and STBout TOs. (**E**) Feature Plots of ADAMTS6 in STBin and STBout TOs. (**F**) RNA FISH of PAPPA2 in STBin and STBout TOs. (**G**) RNA FISH of ADAMTS6 in STBin and STBout TOs. (**H**) DEseq2 was used to find DEGs between STBin and STBout datasets in STB-1, STB-2, and STB-3 clusters and plotted as a volcano plot. GO terms associated with DEGs in either STBin or STBout were found with clusterProfiler and colored on the UMAP with select representative genes highlighted with text. (**I**) Feature Plot of representative target genes from E.

The online version of this article includes the following figure supplement(s) for figure 3:

*Figure 3 continued on next page*

**Figure supplement 1.** STB subtype analysis and expression differences in STBin vs STBout TOs.

**Figure supplement 2.** CTB subtype analysis and expression differences in STBin vs STBout TOs.

**Figure supplement 3.** Representative images of RNA FISH in TOs.

functions. Of note, STB-1 and STB-2 nuclear subtypes were enriched in STB$^{in}$ TOs while STB-3 subtype was doubled in STB$^{out}$ TOs, indicating STB nuclear subtype proportions are sensitive to the culture conditions TOs are grown in (*Figure 3A*).

Given the different proportions of STB subtypes between STB$^{in}$ and STB$^{out}$ TOs, we next sought to determine if gene expression changed within each STB subtype as a function of culture condition. Therefore, we performed pseudobulk differential expression analysis using DESeq2 to compare gene expression of each STB subtype between STB$^{in}$ and STB$^{out}$ TOs and determined GO terms associated with the differentially expressed genes (DEGs; *Figure 3H* and *Supplementary file 10*; *Love et al., 2014*). STB-1 and –2 of STB$^{out}$ TOs were enriched for genes involved in pregnancy including genes in the human placenta lactogen family (CSH1 and CSHL1), growth hormone 2 (GH2), the STB-specific gene ENDOU (*Haider et al., 2018*), and the androgen receptor (AR; *Figure 3H–I*). In addition, STB in STB$^{out}$ TOs exhibited increased expression of genes involved in GTPase signaling, including the RhoA GAPs STARD13 and GRAF3 (ARHGAP42) (*Bai et al., 2013*; *Ching et al., 2003*; *Figure 3H–I*). All three STB subtypes in STB$^{out}$ TOs exhibited an enrichment in transport-associated proteins, consistent with the STB being proximal to media (*Figure 3H*, *Figure 3—figure supplement 1E*). In contrast, STB$^{in}$ TOs were enriched in extracellular matrix organization genes but their expression was not specific to STB subtypes (*Figure 3H*, *Figure 3—figure supplement 1F*). The STB-2 subtype in STB$^{in}$ TOs had increased expression of the hypoxia-associated genes LIMD1, HILPDA, and HIF1A as well as the angiogenesis-associated proteins VEGFA and FLT1 (*Figure 3H*, *Figure 3—figure supplement 1G*; *Foxler et al., 2018*; *de la Rosa Rodriguez and Kersten, 2020*; *Shibuya, 2011*), consistent with the increased proportion of the oxygen-sensing STB-2 subtype in STB$^{in}$. Together, these results demonstrate that there are at least three subpopulations of STB in TOs, which differ both in their relative proportions and transcriptional signature between culture conditions.

## Pseudotime and gene network analysis reveals gene regulators in STB$^{out}$ TOs, including the chromatin remodeler RYBP

To assess whether distinct nuclear subtypes represented an STB differentiation trajectory in TOs, we next performed pseudotime gene expression inference using the integrated STB$^{in}$ and STB$^{out}$ datasets described above (*Figure 2D*). We utilized the Slingshot algorithm to establish the global lineage structure for each dataset using undefined starting and ending clusters (*Street et al., 2018*). Following this, we depicted the pseudotime progression on the UMAP plot for the STB lineage (*Figure 4A*). This visualization revealed a continuous trajectory starting from CTB-p subtype, traversing through CTBs to CTB-pf, and progressing through STB-1 and STB-2 before culminating in the STB-3 subtype (*Figure 4A*). To find genes associated with pseudotime, we performed an association test with tradeSeq and found that STB differentiation was marked by an increase in well-known STB marker genes, including ADAM12, PLAC4, and PSG6 (*Figure 4B*; *Aghababaei et al., 2015*; *Chen et al., 2022*; *Moore et al., 2021*; *Tuohey et al., 2013*). Next, we conducted a comparative pseudotime expression analysis between STB$^{in}$ and STB$^{out}$ conditions. Our findings revealed enrichment of several STB-associated genes in STB$^{out}$, such as the secreted metallopeptidase protein ADAM12, the progesterone synthesis enzyme CYP11A1, and the proteoglycan synthesis gene MAN1A2 (*Figure 4C*; *Aghababaei et al., 2015*; *Zhu et al., 2023*). Interestingly, the chromatin remodeler (CR) RYBP and transcriptional activator AFF1 were two genes most significantly associated with the CTB to STB pseudotime trajectory and enhanced in STB$^{out}$ TOs (*Figure 4B–C*), suggesting they may play roles in initiating transcription of genes involved in STB$^{out}$ nuclear differentiation.

To obtain a global view of candidate TFs and CRs involved in STB differentiation in tissue, STB$^{in}$, and STB$^{out}$ TOs we next turned to a gene network analysis approach. We isolated the subset of nucleus types involved in STB differentiation (CTB, CTB-pf, and STBs) from each dataset and performed RNA velocity using spliced/unspliced matrices and plotted trajectories from each on a UMAP integrated for replicates of each sample type. This analysis creates a trajectory like slingshot but instead of

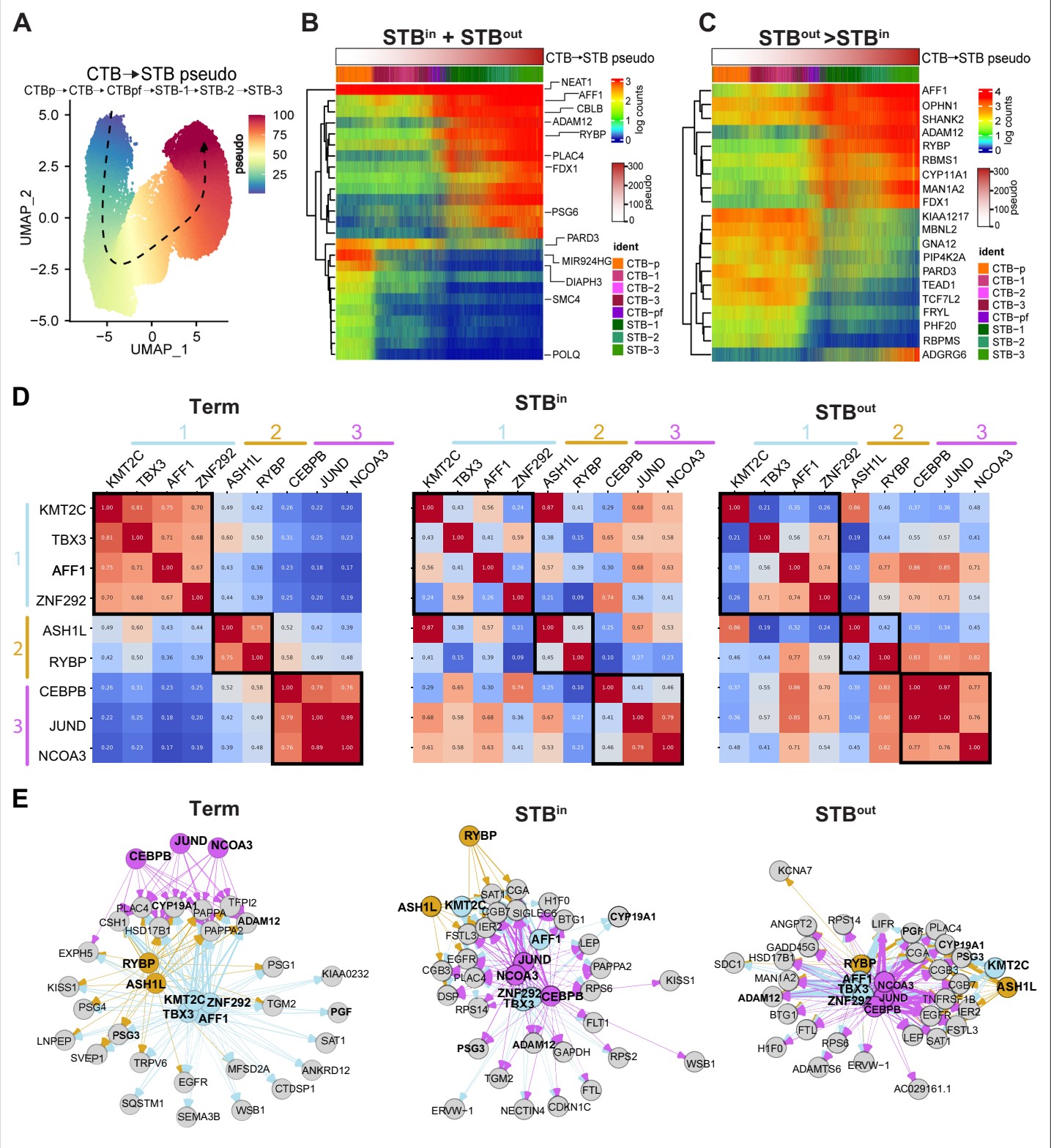

**Figure 4.** Trajectory and gene regulatory network analysis reveals transcription factors and chromatin remodelers associated with STB differentiation. (**A**) Pseudotime was performed with Slingshot and assigned pseudotime value for each nucleus type is overlaid as a color scale on the UMAP from time 0 (blue) to time 100 (red). Arrow demonstrates directionality of the trajectory. Top genes associated with pseudotime (**B**) or enriched in STB[out] compared to STB[in] (**C**) are plotted on a barplot with nuclei ordered with the pseudotime seen in A. Logcounts and pseudotime are demonstrated as a color scale bar and the identity of each nucleus is labeled with a color as indicated in the legend. (**D**) TF/CR and TG pairs were identified via Velorama and plotted

*Figure 4 continued on next page*

*Figure 4 continued*

as a similarity matrix to show the overlap in TGs for each TF/CR. Each module discussed in the text is labeled as 1, 2, or 3 and colored with blue, orange, or pink, respectively. (**E**) Network graphs of TF/CRs and their respective TGs were plotted with igraph. Each TF/CR and respective arrows were false colored as per the module labels in D. Grey circles represent the target genes. Target genes that are shared among all three sample types are bolded. The width of the arrow from each TF/CR to each TG represents the interaction strength score determined with Velorama and the color of the arrow represents the module from which the interaction arises.

The online version of this article includes the following figure supplement(s) for figure 4:

**Figure supplement 1.** RNA velocity traces.

accounting for pseudotime with bulk RNA expression, it leverages transcript splicing dynamics (*La Manno et al., 2018*). The predominance of STB nuclei in the SN full-term tissue dataset precluded the ability to attain sufficient CTB-pf nuclei, but the velocity map demonstrated CTB differentiating into two STB lineages (*Figure 4—figure supplement 1*, **full-term**). Given the decreased STB:CTB ratios present in organoids, we were able to capture high numbers of CTB-pf nuclei and observed that this population is a precursor to the STB, as anticipated (*Figure 4—figure supplement 1*, **STB**$^{in}$ **and STB**$^{out}$). We then employed Velorama to infer gene regulation in these cells. Velorama is an RNA Velocity-based causal inference method that accounts for the multi-trajectory development of cellular state. It infers temporal causality on a directed acyclic graph where each node is a cell, and the edges indicate the velocity-implied direction of differentiation. Velorama derives gene regulatory networks by training a neural network to predict target gene (TG) expression profiles given regulator genes, such as TFs, and outputs an interaction score for every TF and TG combination (*Singh et al., 2024*). Given the identification of the CR RYBP in the Slingshot analysis, we expanded Velorama to include other known CRs, as they also have the potential to regulate gene expression levels.

To dissect the functional interactions between TF/CRs, we first compared the similarity of TGs interacting with each TF/CR and plotted this as a heatmap with a score of 1 indicating all TGs are shared and a score of 0 indicating no overlap (heatmap demonstrates high overlap in red and low overlap in blue, *Figure 4D*). In full-term tissue, these TF/CR fell into three modules that shared most TGs: 1- KMT2C/TBX3/AFF1/ZNF292 (blue), 2-ASH1L/RYBP (orange), and 3-CEBPB/JUND/NCOA3 (pink; *Figure 4D*). In contrast to full-term tissue, most TGs in STB$^{in}$ organoids were associated with AFF1/JUND/NOCA3/ZNF292/TBX3/CEBPB in no module order (*Figure 4D*). Finally, STB$^{out}$ appeared to have a similar distribution of TF/CRs as STB$^{in}$, apart from an increased prevalence of RYBP association with TGs in STB$^{out}$ (*Figure 4D*).

We next evaluated the specific TGs in each condition to predict the possible functions of the TF/CR and TG pairs. In full-term tissue, many STB-specific TGs were found to be associated using RNA velocity including CYP19A1, MFSD2A, ADAM12, PSGs, human placenta lactogen (CSH1), and the placenta specific insulin regulator PAPPA enriched in STB of full-term tissue (*Wang et al., 2024*). Module 3 was only associated with a handful of genes at full-term and with low interaction scores (interaction score demonstrated by arrow width). In contrast, Module 3 had a much stronger prevalence in both the number of TGs and interaction scores in both STB$^{in}$ and STB$^{out}$ organoids with some similar genes (conserved genes bolded, ADAM12, CYP19A1, PGF, PSG3) and some unique to organoids, particularly of note being hCG genes (CGA, CGB4, CGB7; *Figure 4E*), whose expression is known to be decreased in the STB of full-term tissue (*Rull and Laan, 2005*). STB$^{in}$ was the only condition to show a link between the TF/CRs and the VEGF receptor FLT1, consistent with the enrichment of hypoxic/angiogenic associated genes in the STB-2 subtype of STB$^{in}$ compared to STB$^{out}$ (*Figures 3H and 4D*). Importantly, this analysis independently identified both RYBP and AFF1 as transcription effectors involved in the expression of STB TGs, consistent with our pseudotime analysis (*Figure 4A–C*). AFF1 was associated with most TGs in all three sample types (*Figure 4E*). In contrast, while RYBP was associated with most STB marker genes in full-term tissue and STB$^{out}$, it was associated with only a few TGs in STB$^{in}$ (*Figure 4E*). In fact, RYBP was associated with genes that were enriched in STB$^{out}$ organoids in the Slingshot analysis (ADAM12 and MAN1A2) but not in STB$^{in}$ TOs (*Figure 4B–E*), supporting the hypothesis that RYBP could be a modulator of a specific lineage of STB nuclear subtype differentiation. In summary, this gene network analysis suggests that many TF/CR and TG interactions are shared amongst tissue and TOs. However, many interactions in TOs also change as a function of culture condition. While module 3 is most linked to TGs expressed in TOs, RYBP is linked specifically to TGs expressed in tissue and STB$^{out}$ TOs, but not STB$^{in}$ TOs.

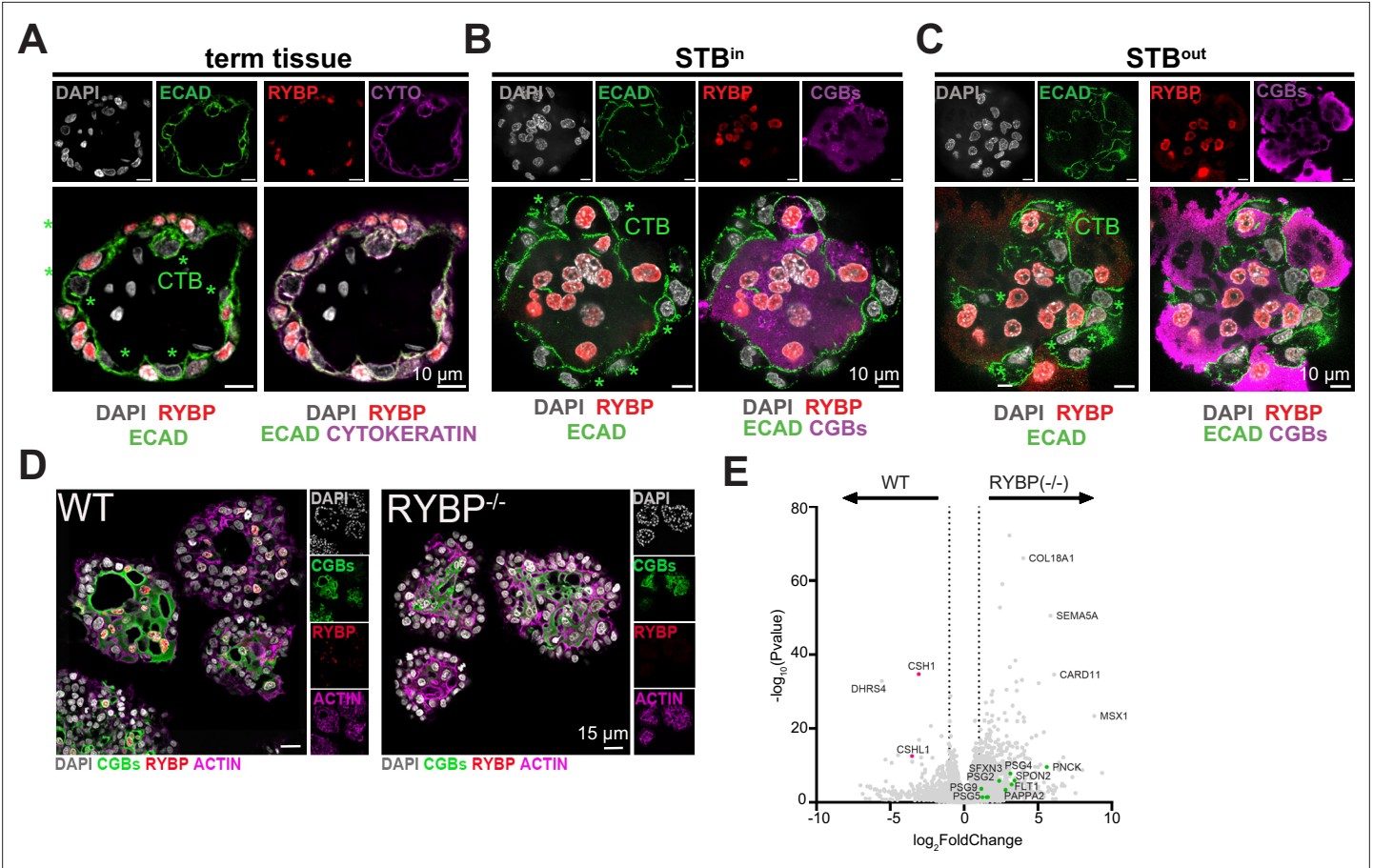

**Figure 5.** RYBP is a marker of STB and its deletion downregulates STB-2 marker genes. (**A**) Immunofluorescence of full-term tissue was performed with E-cadherin (ECAD), cytokeratin (CYTO), and RYBP and stained with DAPI to label nuclei. Immunofluorescence of STB[in] TOs (**B**) or STB[out] TOs (**C**) was performed with chorionic gonadotropin (CGBs) to label STB, RYBP, and E-cadherin (ECAD) to mark cell boundaries. TOs were further stained with DAPI to label nuclei. Green stars represent CTB cells, as marked by E-cadherin, and are negative for RYBP. (**D**) Immunofluorescence of WT or RYBP-/- STB[in] TOs performed with RYBP and CGBs and stained with DAPI and Actin. (**E**) Bulk RNA sequencing of WT or RYBP(-/-) STB[in] TOs was performed and analyzed with DESeq2 to determine differential gene expression. Dotted line represents a log$_2$ fold change greater than 1. Genes enriched in the STB-2 subtype are labeled in green.

The online version of this article includes the following figure supplement(s) for figure 5:

**Figure supplement 1.** The STB marker ENDOU demonstrates E-cadherin and cytokeratin-7 can designate CTB and STB nuclei in tissue.

**Figure supplement 2.** Sanger sequencing validation of RYBP and AFF1 KO TO clones.

**Figure supplement 3.** The deletion of AFF1 in TOs results in the downregulation of STB-2 marker genes.

## RYBP is a STB-specific nuclear marker involved in silencing gene expression characteristic of the STB-2 nuclear subtype

To validate the analyses described above, we selected RYBP as a model TG for its potential role in STB nuclear subtype differentiation and based on the availability of well-validated antibody reagents. RYBP contains chromatin remodeling activity as part of the PRC1 histone ubiquitination complex and can both increase and decrease the transcription of TGs (*Rose et al., 2016*; *Simoes da Silva et al., 2018*). The expression dynamics of RYBP was directly correlated with the differentiation trajectory from CTB to STB, with upregulated expression beginning in CTB-pf (*Figure 4B*). We first visualized RYBP in full-term tissue sections, using E-cadherin as a marker of CTB and Cytokeratin-7 as a marker of CTB and STB cells. STB can be clearly designated as regions expressing Cytokeratin-7 but not expressing E-Cadherin, as validated with the STB-specific marker ENDOU (*Figure 5—figure supplement 1*). We found that RYBP exhibited STB-specific localization in term placenta tissue (*Figure 5A*). To confirm this result in TOs, we visualized RYBP at the protein level in each sample type with immunofluoresence (IF)

using E-cadherin as a CTB marker and CGBs as STB specific markers (*Figure 5B–C*). RYBP exhibited STB specific expression in both STB$^{in}$ and STB$^{out}$ TOs. Together this suggests that RYBP is a novel marker of STB nuclei in tissue and TOs. We hypothesized that RYBP could play a role in guiding STB nuclei toward a specific nuclear subtype during differentiation. To test this, we performed gene editing in TOs with CRISPR/Cas9 to delete RYBP and confirmed this knock-out by sequencing and IF (*Figure 5D*, *Figure 5—figure supplement 2A*). In parallel, we generated TOs with AFF1 deletion, as it was also associated with the STB differentiation trajectory and confirmed the knockout by sequencing (*Figure 4*, *Figure 5—figure supplement 2B*). However, despite testing several antibodies, none were suitable for IF localization of AFF1. We generated three homozygous knockouts clones of each line (RYBP$^{-/-}$) or AFF1 ($^{-/-}$) and compared their morphologies to three control TO lines that contained the Cas9 plasmid but a scrambled gRNA (WT). No difference in the overall morphology of TOs was found, and size and number of nuclei within the STB remained constant (*Figure 5D*). To determine if gene expression differences were found in the TOs lacking RYBP or AFF1, we performed bulk RNA sequencing of each RYBP($^{-/-}$), AFF1 ($^{-/-}$) or WT lines in STB$^{in}$ TOs. We found that deletion of either RYBP or AFF1 resulted in significant increased expression of genes associated with the STB-2 oxygen sensing subtype, including FLT1, PAPPA2, SPON2, SFXN3, and many of the PSG hormones involved in angiogenesis (*Figure 5E*, *Figure 5—figure supplement 3*, *Supplementary file 11*). RYBP($^{-/-}$) TOs upregulated transcripts that were not expressed in WT TOs or AFF1$^{-/-}$ TOs (*Figure 5E* and *Figure 5—figure supplement 3*). RYBP is a component of a non-canonical Polycomb complex that transcriptionally silences genes (*Rose et al., 2016*). In addition to the upregulation of the STB-2 genes, its knockout led to the upregulation of many genes not typically expressed in TOs including COL18A1, SEMA5A, CARD11, and MSX1, likely due to the loss of this silencing function (*Figure 5E*). In contrast, the human placenta lactogen gene (CSH1) was significantly upregulated in WT TOs compared to RYBP$^{-/-}$ and AFF1$^{-/-}$ TOs, suggesting deletion of both RYBP and AFF1 affect the expression of a hormone increased in STB$^{out}$ TOs (*Figure 5E*, *Figure 5—figure supplement 3*, and *Supplementary file 11*). However, no further STB-3 subtype genes were found to be upregulated compared to control in the STB$^{in}$ TOs tested. Together, these results validate the predictions from the trajectory analyses described above and suggest that AFF1 and RYBP act to silence the STB-2 nuclear subtype and activate human placenta lactogen in TOs.

## Comparison of STB differentiation in TOs to first trimester and full-term placental tissue

Having identified several differences in STB differentiation between STB$^{in}$ and STB$^{out}$ TOs, we next sought to identify how these differences relate to placenta tissue across gestation. We used a publicly available first trimester SN sequencing dataset and integrated this data with the full-term tissue and TOs described above (*Figure 6—figure supplement 1A*). Because tissue-derived datasets contain a large proportion of non-trophoblast nucleus types, we subset the integrated dataset to contain only trophoblasts in the STB lineage, which included both proliferating CTBs (CTB-p), CTBs (CTB-1 and CTB-2), pre-fusion CTBs (CTB-pf), and five distinct STB populations (*Figure 6A, B*, *Figure 6—figure supplement 1A-D*). The proportion of STB:CTB was higher in full-term tissue than in first trimester tissue (7:1 at full-term vs 1.5:1 in first trimester, combined dataset), as anticipated given the higher proportion of CTBs in first trimester (*Figure 6B*; *Benirschke et al., 2012*; *Mayhew, 2014*). Of note, the STB:CTB ratios in TOs more closely resembled first trimester than term tissue in both culture conditions (1:1 for STB$^{out}$ and 0.6:1 for STB$^{in}$; *Figure 6B*). The tissue and TO samples exhibited dramatically different proportions of each CTB (CTB-1–2) subtype and each STB (STB1-5) subtype, which indicates sample to sample heterogeneity (*Figure 6A, B*, *Figure 6—figure supplement 1A-C*). Therefore, we merged the two CTB and five STB populations together to globally detect the conserved and DEGs among the STB or CTB populations (*Figure 6C*). We first analyzed the expression of canonical marker genes for each nucleus type (for example TP63/TENM3 in CTB and CYP19A1/TFAP2 A in the STB) and found that each sample expressed these markers (*Figure 6D*), suggesting general conservation across both tissue types and TO conditions. To dissect what was different amongst the samples, we next analyzed the enriched genes in each nucleus type and identified GO terms associated with these genes for each sample type (*Figure 6E* and *Figure 6—figure supplement 1E* and *Supplementary files 12 and 13*). GO terms associated with the CTB populations were remarkably conserved amongst all sample types (*Figure 6—figure supplement 1*). In contrast, STB exhibited greater diversity of

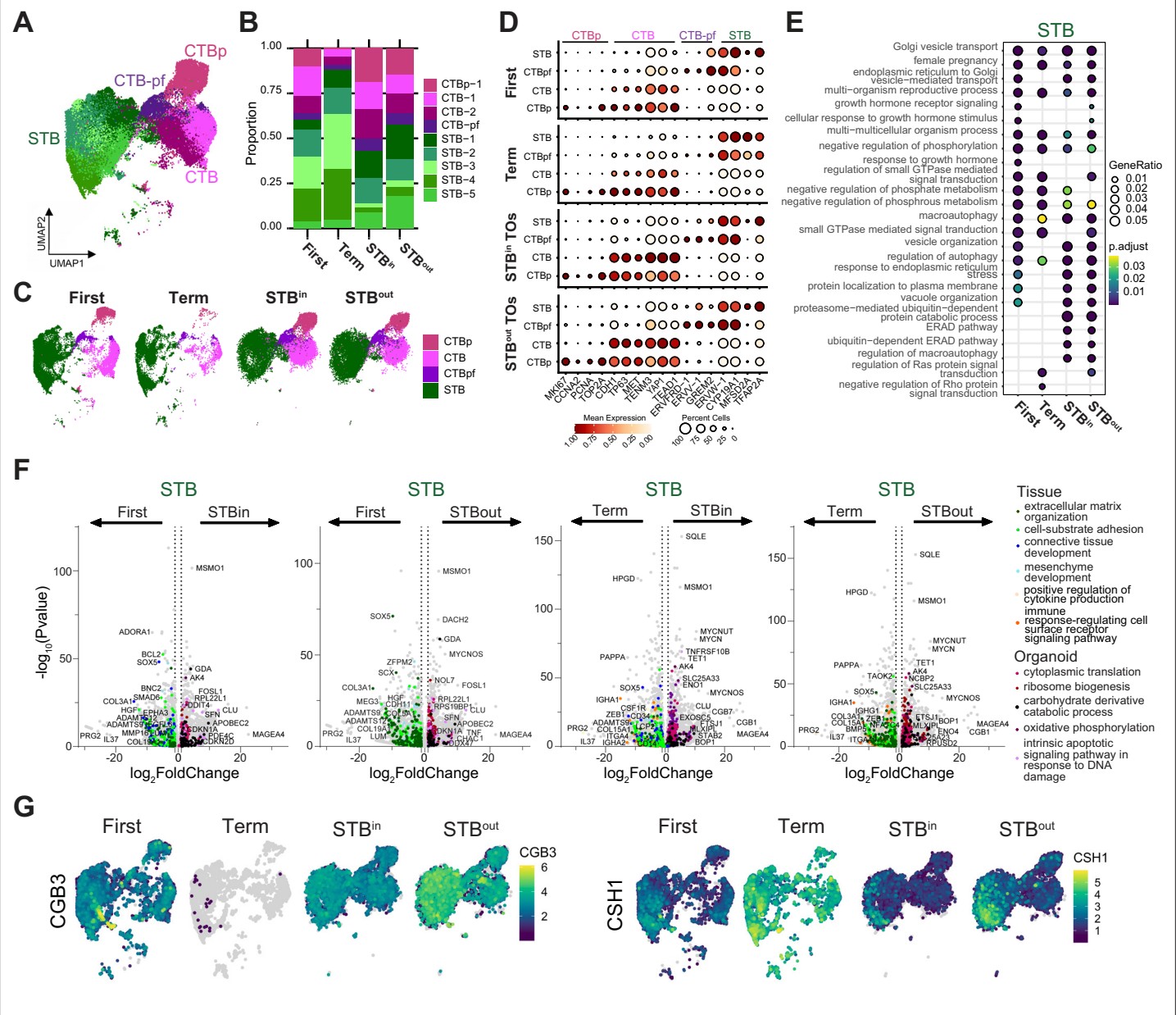

**Figure 6.** Similarities and differences between STB in first trimester and full-term tissue and TOs. (**A**) UMAP of the integrated SN dataset including first trimester, full-term tissue, STBin TOs, and STBout TOs. (**B**) Relative proportion of each nucleus type for each sample in the integrated dataset in A. (**C**) UMAP of the integrated dataset split by sample type. The five STB subtypes were merged into a single STB subtype while the two non-proliferative CTB subtypes were merged. (**D**) Dot plot of key trophoblast marker genes in the STB lineage with heatmap representing mean gene expression and size of the dot representing the percent of nuclei expressing each marker. The labels on the top of the graph refer to the established marker genes used to identify the identity of each cluster. (**E**) The top DEG in the STB of each sample type were analyzed for biological process GO term enrichment with clusterProfiler and plotted as a dotplot. (**F**) DEseq2 was used to find DEGs between different sample types for the merged STB cluster. GO terms for DEGs enriched in each sample type were found with ClusterProfiler and colored on the UMAP with select representative genes highlighted in text. (**G**) Feature plots demonstrating RNA expression of selected genes.

The online version of this article includes the following figure supplement(s) for figure 6:

**Figure supplement 1.** Characterization of nucleus types in merged SN dataset with TOs, first trimester, and full-term tissue.

functions between sample types (*Figure 6E*). While GO terms involved in ER to Golgi transport and hormone production were conserved in all STB samples, many processes were found only within a subset of sample types (*Figure 6E*). For example, genes with ER stress, ER associated degradation, and macroautophagy GO terms were expressed in TOs but not in tissue. In contrast, GTPase

signaling and vesicle organization were present in first trimester, term, and STB^out TOs, but not in STB^in (*Figure 6E*). Together, these findings suggest that the STB exhibits greater heterogeneity in function across sample types than do CTBs.

To determine what genes were significantly different in each sample, we performed pseudobulk analyses using DESeq2, plotted genes on a Volcano plot, and colored genes significantly associated with different GO terms (*Figure 6F* and *Supplementary file 14*). We performed this analysis comparing STB in either first trimester tissue to TOs (STB^out or STB^in) or full-term tissue to TOs (STB^out or STB^in) (*Figure 6F*). Many GO terms were enriched across all comparisons. This included an increase in extracellular matrix, cell substrate adhesion, and connective tissue development genes within the tissue datasets, with a particular enrichment of collagen genes and matrix metalloproteases (ADAM and MMP family members; *Figure 6F*; *Qu and Khalil, 2022*). In contrast, the STB present in organoids exhibited an enrichment of genes involved in cytoplasmic translation/ ribosome biogenesis as well as oxidative phosphorylation/ DNA damage (*Figure 6F*). In addition, genes associated with cytokine production and the immune response based on GO terms were present in full-term tissue but not first trimester tissue or TOs (*Figure 6F*). Therefore, while STB of TOs and tissue share many conserved marker genes, a subset of STB gene expression is differentially regulated between tissue and TOs.

We next evaluated the expression of genes known to change throughout gestation in each sample type. For example, many hormones are differentially released throughout pregnancy with first trimester tissue exhibiting high hCG production (CGB genes) and full-term tissue releasing more human placenta lactogen (CSH genes; *Costa, 2016*; *Rull and Laan, 2005*; *Samaan et al., 1966*). Consistent with these gestational hormone trends, STB from first trimester expressed hCG genes while there was no expression in full-term tissue (*Figure 6G*, *Figure 6—figure supplement 1F*). Further, full-term tissue exhibited an increase in CSH1 expression compared to first trimester (*Figure 6G*, *Figure 6—figure supplement 1F*). STB^in and STB^out TOs expressed both hormones in the STB suggesting organoids can produce hormones differentially expressed throughout gestation (*Figure 6G*, *Figure 6—figure supplement 1F*). STB^out TOs exhibited an increase in expression of both hormones compared to STB^in TOs (*Figure 6G*, *Figure 6—figure supplement 1F*). Consistent with previous reports, there was an increase in the expression of FLT1 and decrease in PAPPA in the STB of first trimester compared to full-term tissue, with TOs resembling first trimester expression (*Figure 6—figure supplement 1G–H*; *Wang et al., 2024*). Together, these results demonstrate that while the STB of each sample expresses key STB marker genes, both gestational age and TO culture conditions can modify the gene expression patterns of the STB.

## Discussion

A critical barrier to understanding human gestation has been the limited number of accessible models for the placenta. In this study, we conducted comparative SC and SN RNA sequencing on full-term placenta tissue and TOs. Our findings demonstrate that SN sequencing is crucial for capturing the STB lineage due to its distinct syncytial structure. In contrast, SC sequencing enriched for mitotic cell populations in TOs and EVT in tissue, suggesting the isolation approach chosen depends on your trophoblast cell type of interest. We characterized the nucleus types in each TO model and utilized DEG and pseudotime analyses to define three STB subtypes present in both STB^in and STB^out TOs, albeit at different ratios. These include a juvenile population that exhibited intermediate CTB and STB expression (STB-1), an FLT1 +population enriched in genes involved in oxygen sensing and the stress response (STB-2), and a final subtype enriched in transport and GTPase signaling molecules (STB-3). We identified the CR RYBP as a gene linked to STB differentiation. We validated that RYBP exhibits STB-specific expression in TOs and tissue via immunofluorescence and utilized CRISPR to knock out RYBP in TOs. Deletion of RYBP in TOs upregulated genes that define the STB-2 subtype and downregulated the placenta hormone human placenta lactogen. Finally, we compared STB gene expression between placental tissues from first trimester and full-term to TOs. This showed that although standard STB differentiation markers are maintained in all, there is substantial heterogeneity in STB gene expression between the different sample types. Together, these results draw important implications for our understanding of STB nuclear differentiation and show how TOs can serve as relevant STB models.

STB^out TOs grown in suspension maintain native polarity and exhibit an increase in syncytia size, with >50 nuclei/syncytia in STB^out compared to ~10 nuclei/syncytia under standard STB^in culture

conditions (*Yang et al., 2024*). However, it was unknown whether CTB grown via this method maintained their proliferative capacity. Here we show that the proportion of mitotic cells remains constant between TOs grown in STB^in and STB^out conditions, suggesting culture in suspension is not a terminal, post-mitotic state. In fact, the proportion of STB nuclei increased in the STB^out condition was concurrent with a decrease in the CTB-CC population. In vivo, CTB-CC cells sit adjacent to the villous trees and maternal uterus and are thought to progressively differentiate into EVT cells that then can invade into the uterus (*Arutyunyan et al., 2023*; *Boyd and Hamilton, 1970*; *Turco and Moffett, 2019*). Therefore, the increase of STB in the STB^out condition could be caused by promotion of STB lineage and an inhibition of differentiation down a CTB-CC or EVT lineage.

The enrichment of STB nuclei in SN sequencing allowed us to identify and define three populations of STB in the STB^in and STB^out TO conditions. STB-1 represented a juvenile population undergoing a transition from CTB to STB gene expression. These nuclei may have recently incorporated into the syncytia and are actively undergoing differentiation at the time of sequencing. STB-2 expressed the VEGF receptor FLT1 and is enriched in genes involved in responding to oxygen levels and ER stress, suggesting STB-2 might play a role in responding to low oxygen levels. Finally, STB-3 exhibits an increase in GTPase signaling molecules and transporter proteins. Importantly, culture conditions changed the relative enrichment of these subtypes. Whereas STB^in TOs were enriched for the juvenile STB-1 and oxygen sensing STB-2 populations, STB^out TOs exhibited an increase in the transport/GTPase STB-3 subtype. We hypothesize the STB-3 subtype is a terminal differentiation state as it appears at the end of the slingshot pseudotime trajectory. In fact, in addition to exhibiting a higher proportion of the STB-3 subtype, STB^out TOs express higher concentrations of key STB pregnancy hormones in all nuclear subtypes suggesting culturing in suspension may promote terminal STB differentiation. An alternative hypothesis, however, is that nuclei in STB^in and STB^out TOs exhibit distinct lineages trajectories. Testing these hypotheses will require future experiments tracing nuclear subtypes through time.

Why might changing culture conditions influence the distribution and gene expression of STB nuclear subtypes? We hypothesize growth in either extracellular matrix (STB^in) or suspension (STB^out) impacts both the cell orientation and environmental cues each cell type is exposed to like oxygen concentrations, cell and membrane tension, and media flow dynamics. Future studies dissecting each environmental cue and the differentiation of STB nuclei through time and space will help elucidate the molecular mechanism driving each STB subtype. Of note, these three STB subtypes are remarkably like those recently defined in first trimester and full-term tissue: a juvenile population, a FLT1 expressing population enriched in genes involved in oxygen sensing, and a PAPPA positive population that expresses GTPase signaling molecules and hormones (*Wang et al., 2024*). The distribution of STB subtypes defined in *Wang et al., 2024* also changed throughout gestation with the oxygen sensing STB nuclear subtype increased in the first trimester and the GTPase/hormone subtype increased at term and implies the environmental cues the STB experiences throughout gestation may affect STB subtype distribution in vivo similar to what is seen in TOs. This resemblance suggests TOs can recapitulate similar nucleus subtypes as those seen in vivo, indicating their strength as an experimental model and highlights the importance of dissecting the environmental cues that contribute to each STB subtype in vitro.

How might these distinct nuclear transcriptional identities affect the function of the STB? Despite the STB being one large cytoplasm where molecules can freely mix by diffusion, it has long been suggested to contain distinct cytoplasmic zones that specialize in different functions (*Benirschke et al., 2012*; *Burgos and Rodriguez, 1966*; *Burton, 1990*). For example, STB cytoplasmic regions adjacent to the fetal vasculature dramatically thin presumably to facilitate diffusional exchange of gas and nutrients and express angiogenic promoting proteins. In contrast, regions where hormones are produced have dense packing of membraneous organelles and express protein trafficking molecules (*Baczyk et al., 2004*; *Beck et al., 1986*; *Burton and Jauniaux, 1995*; *Clark et al., 1998*; *Hempstock et al., 2003*; *Jauniaux et al., 2003*; *Khaliq et al., 1996*; *Morrish and Marusyk, 1997*; *Hauguel-de Mouzon, 1997*; *Sharkey et al., 1993*). One potential mechanism for creating these distinct cytoplasmic zones is nuclear specialization, whereby individual nucleus identities might be spatially localized to different cytoplasmic regions, as has been demonstrated for nuclei in syncytial muscle fibers (*Kim et al., 2020*; *Petrany et al., 2020*). Different nuclear identities likely arise from a combination of differentiation pathways and environmental cues. The generation of an organoid cell culture model

that recapitulates the STB nuclear identities seen in vivo is a critical advancement towards deciphering the mechanisms that allow the giant STB cell to effectively carry out its many essential functions.

To dissect the TFs that drive STB differentiation we applied gene regulatory analysis and identified multiple TF/CR modules potentially involved in differentiation of STB in full-term tissue and TOs. Interestingly, the module most associated with STB[in] included the TF CEBPB, which is important for placenta development in mice and was recently found to be involved in STB differentiation in first trimester but not in full-term tissue (*Bégay et al., 2004*; *Wang et al., 2024*). Consistent with these results, we found that the TF CEBPB was only associated with a few genes in our independent full-term tissue dataset, suggesting that STB of TOs might employ transcriptional programs characteristic of the first trimester. The final module includes the CR RYBP, which is involved in a non-canonical form of the PRC1 polycomb complex that ubiquitinylates histones and can modulate gene expression (*Rose et al., 2016*; *Simoes da Silva et al., 2018*). We demonstrate that RYBP exhibits STB-specific expression in TOs and tissue. We predicted that RYBP might mediate the differentiation of STB nuclei into specifc nuclear subtypes. To test this idea, we deleted RYBP with CRISPR/Cas9 in TOs and observed an increase in many STB-2 nuclear subtype marker genes and a significant decrease in the human placenta lactogen gene. This suggests that RYBP acts to downregulate the STB-2 nuclear subtype in STB[in] TOs. Future work delving into the exact distribution of STB nuclear subtypes in STB[in] and STB[out] TOs will enable us to dissect if RYBP also plays a role in activating the STB-3 subtype, or simply inhibiting the STB-2 subtype. Of note, RYBP deletion in mice is embryonic lethal due in part to a failure to form trophectoderm and subsequent invasion defects (*Pirity et al., 2005*), consistent with a possible role of RYBP on STB differentiation in human placenta.

While the global subtypes found in first trimester, full-term tissue, STB[in] TOs, and STB[out] TOs were similar, many genes were differentially expressed in each STB population. One significant difference between the STB of full-term tissue and TOs was in hormone expression. It has long been appreciated that the STB differentially expresses hormones as a function of gestational age (*Costa, 2016*; *Kumar and Magon, 2012*), but it was not known 1- how TOs mirror this expression and 2- how isolation of TOs from different stages of tissue gestation affected expression. We found that while the TOs used in this study were derived from full-term CTBs, the STB associated with these organoids express hormone transcripts classically associated with early gestation. This suggests that the cues that restrict these hormones to different gestational stages in vivo are not intrinsic to the isolated CTBs and can be studied in full-term derived TOs. Future work adding maternal cues to the TO system will help define how STB hormone levels are mechanistically modulated.

In conclusion, our study elucidates STB nucleus subtypes in TOs, tracks their proportions across culture conditions, and compares gene expression to STB in vivo. The fluctuating proportions of STB subtypes across TO culture conditions imply that environmental cues can direct individual nuclei in the same cell into different identities. These findings underscore the power of TOs as an experimental model for studying the STB.

## Methods
### Tissue processing for SC and SN sequencing

Placenta tissue was collected from patients undergoing scheduled C-sections at UNC Health consented under IRB 21–2055. Inclusion criteria included patients undergoing scheduled C-sections at UNC Health over 18 years of age. Patients were approached in clinic during routine prenatal care. After explaining the study, reviewing the informed consent form, and answering any questions, the patient and consenter signed and dated the consent form. The signed consent form included approval for genomic studies, derivation of cell lines, and publication of results. Patient information, sequencing data, and tissue samples from these experiments was later transferred to Duke under the IRB Pro00113088. Immediately after placenta delivery a cotyledon from the center of the placenta was dissected. Samples from both decidua and villous tissue were snap frozen in liquid nitrogen for future processing within ten minutes of placenta delivery to minimize STB degradation. Additional samples were fixed in 10% buffered formalin for subsequent tissue paraffin embedding and slicing. The remaining villous/decidua tissue from the dissected cotyledon was then immediately processed into single cells, as described below. A list of the tissues used in this study is available in *Supplementary file 1*.

## SC processing

The cotyledon (decidua +villous, chorion removed) was chopped into fine pieces (<1 mm) with a scalpel and washed with 1 x PBS in cheese cloth until flow through was clear of blood. Tissue was placed into Trypsin media (1 X PBS without calcium or magnesium, 0.2% Trypsin (Thermo 15090046), 0.53 M EDTA) and incubated at 37 °C for ten minutes in a shaking water bath. After incubation trypsin was inactivated with 100mLs of Wash Media (DMEM F12 media +20% FBS). Supernatant was passed through a sterile cheese cloth, spun down, and resuspended to Resuspension Media (DMEM F12 media +10% FBS). Remaining tissue was then placed into 25 mL collagenase buffer (1 mg/mL collagenase V (Sigma C9263) in Wash Media) and shaken at 37 °C for 10 min. Supernatant was passed through cheese cloth, spun down, and resuspended in Resuspension Media. Cell pellets from both digestion steps were combined and pelleted, resuspension media removed, and resuspended in 10mLs RBC lysis buffer (Thermo 00-4333-57) and incubated at RT for 10 min. Cells were passed through a 100 μm filter, spun down, and passed through a Milltenyi Debris Removal Solution (130-109-398) gradient as per the manufacturer's instructions. The final pellet was then resuspended in 1 X PBS +0.04% BSA (Sigma A1595).

## SN processing

Single nuclei were isolated with the 10 X Chromium Nuclear Isolation Kit (CG000505) as per the User Guide with the following changes. 50mgs of frozen tissue was resuspended in lysis buffer on ice, dounced to homogenize, and incubated for a total of only 7 min from the resuspension step to centrifugation. In addition, while the initial centrifugation step in lysis buffer was performed at 500 x $g$ for 5 min to minimize time spent in lysis buffer, subsequent wash spins were done at 500 x $g$ for 10 min to minimize loss. The final pellet was then resuspended in 1 X PBS +0.04% BSA (Sigma A1595)+10 X Genomics supplied RNAse inhibitor.

## Organoid culture

STB$^{in}$, STB$^{out}$, and EVT$^{enrich}$ TOs were derived, propagated, and differentiated as described previously (*Yang et al., 2022* and *Yang et al., 2024*). Briefly, STB$^{in}$ TOs were derived and maintained by sequential digestion of term placental chorionic villi with 0.2% trypsin-250 (Alfa Aesar, J63993), 0.02% EDTA (Sigma-Aldrich, E9884), and 1.0 mg/mL collagenase V (STEMCELL Technologies, 100–0681), followed by further mechanical disruption by pipetting. Pooled digests were washed with Advanced DMEM/F12 medium (Gibco 12634–010) and pelleted by centrifugation, then resuspended in ice-cold Matrigel (Corning 356231). Matrigel 'domes' (40 μl/well) were plated into 24-well tissue culture plates (Corning 3526) and overlaid with 500 μL prewarmed term trophoblast organoid medium (tTOM; *Supplementary file 3*). Cultures were maintained in 37 °C humidified incubator with 5% $CO_2$. Medium was renewed every 2–3 days. To generate STB$^{out}$ TOs, mature STB$^{in}$ TOs were released from Matrigel domes with cell recovery solution (Corning, 354253) on ice for 30–60 min, pelleted, washed one time with cold basal media (Advanced DMEM/F12+1% P/S+1% L-glutamine +1% HEPES), and then resuspended in pre-warmed tTOM supplemented with 5 μM Y-27632, and transferred into an ultra-low attachment 24-well plate (Corning 3473) for suspension culture at 37°C and 5% $CO_2$ for 48 hours. To generate EVT$^{enrich}$ TOs, established STB$^{in}$ TOs were passaged into new Matrigel 'domes' as described above and previously (*Yang et al., 2022* eLife), and maintained in tTOM for ~5 days prior to switching to EVT differentiation media 1 (EVT m1 recipe: *Supplementary file 4*) for 9 days culture, then replaced with EVT m2 with the same recipe as EVT m1, but lacking NRG1 for a further 3–4 days. A list of the TOs lines used in this study is available in *Supplementary file 1*.

## CRISPR/Cas9-mediated genes editing in TOs

To generate *RYBP* and *AFF1* knock out TOs line, two sets of sgRNA pairs for each gene with the target sites close to the 5 prime of genes ORF were selected from a pre-designed sgRNA database established by Synthego (https://design.synthego.com/#/), then each sgRNA pair (dual-sgRNAs) were cloned into individual cassettes on Lentiviral transfer plasmid with Cas9 ORF (Dual-gRNA lentivirus CRISPR vector). CRISPR lentivirus was produced in 293T cells by the transient transfection of the combination of above genes specific transfer plasmids, psPAX2 (packaging plasmids) and pMD2.G (envelop plasmids) at the ratio 2:1:1 following the standard lentivirus packaging protocols described previously (*Hatterschide et al., 2023*). Using the prepared lentivirus, we transduced the fully dissociated

TO single-cell suspension overnight in an ultra-low attachment culture vessel. The cells were then replated into fresh Matrigel domes for further culture and recovered for recovery 2–4 days prior to puromycin selection (2 µg/ml) for ~4 days. Single organoid unit picking and dissociation for clonal expansion were performed after an additional 2 weeks of growth. Genomic DNA was purified from each individual single-organoid clone for sequencing validation. High-fidelity PCR was conducted on the predicted target region of the gRNAs using the primers listed in *Supplementary file 6*. The purified PCR products were then subjected to Sanger sequencing and analyzed by comparing them to sequences from scramble gRNA-mediated single-organoid clones. Immunofluorescent staining was performed to further validate knockout single-organoid clones. The sequences of the gRNAs used in this study can be found in *Supplementary file 5*.

## Organoid processing for SC and SN sequencing

STB^in TOs were processed into both single cells and nuclei for SC/SN sequencing while STB^out and EVT^enrich TOs were processed into single nuclei for SN sequencing via the following protocols.

### SC processing

STB^in TOs were dissociated by scraping Matrigel domes into 1 mL of pre-warmed TrypLE Express (Invitrogen, 12605036) and incubating at 37 °C for 12 min, swirling the tube every 2–3 min. Dissociated organoids were pelleted at 1250 rpm for 3 min and re-suspended in 200 µL DMEM containing 10% FBS. Resuspended organoids were subjected to vigorous manual disruption using a single channel p200 pipette (Ranin, 17008652) for 3 min followed by the addition of 800 µL of DMEM containing 10% FBS. The disrupted suspension was then passed over a 40 µm filter cell strainer (Corning, 352098). Flow through was then centrifuged at 1250 rpm for 5 min and the pellet resuspended in 250 µL of 1 x PBS for a final volume of ~300 µL and cell counts of ~1 x $10^6$ cells/mL.

### SN processing

TOs from each condition (STB^in, STB^out, EVT^enrich) were harvested by scraping with a wide bore pipette, centrifuged to pellet (600 x *g* for 6 min), resuspended in 100uls TrypLE (Thermo Fisher, 12605010), and incubated at 37 °C for 10 min. After incubation each sample was pipetted 100 x with a P200 pipette to dissociate cells and placed on ice. Single nuclei were then isolated with the 10 x Chromium Nuclear Isolation Kit (CG000505) as per the User Guide with the following changes. 500uls of Lysis buffer was added to the TO/TrypLE solution and transferred to a 2 mL Kimble Dounce (Millipore Sigma, D8938) on ice, dounced 10 x, and subsequently incubated on ice for a total of 10 min. Remaining steps were performed as suggested, except for final wash step spins were performed at 500 x *g* for 10 min to minimize nuclei loss. The final pellet was then resuspended in 10mLs of 1 X PBS +0.04% BSA (Sigma A1595)+10 X Genomiocs supplied RNAse inhibitor, nuclei counted, and 10,000 nuclei run in each well of a chromium controller.

## 10X genomics library generation, sequencing, and data analysis

SC suspensions were stained with Trypan Blue and counted to obtain live/dead cell ratios while SN were stained with Ethidium Homodimer-1 (Thermo E1169) and counted with a hemocytometer on a fluorescent microscope. 10,000 SC/SN of each sample type were loaded into individual chip wells and run on a 10 x chromium controller with the Chromium Single Cell 3' Reagent Reagent Kit v3.1 (Dual Index) following the manufacturers protocol. Tissue libraries were then sequenced on an Illumina NovaSeq S2 at a targeted sequencing depth of 100,000 reads/cell or nucleus while organoid libraries were sequenced on an Illumna NovaSeq S4 at a targeted sequencing depth of 74,000 reads/cell or nucleus. Cell Ranger was then used to align reads to the human genome (GRCh38) and create a counts matrix.

### SC and SN data analysis

Post-processing, quality control, and read alignment to the hg38 human reference genome were performed using 10 x CellRanger package (v6.1.2, 10 x Genomics). Gene expression matrices generated by the 10 x CellRanger aggregate option were analyzed using Seurat (version 4.0) in R (*Butler et al., 2018*; *Hao et al., 2021*; *Satija et al., 2015*; *Stuart et al., 2019*). For SC, cells with at least 200 and no more than 10,000 unique expressed genes were included in downstream analysis, and cells

with more than 25% mitochondrial reads were excluded from analysis. For SN, nuclei with at least 800 and no more than 10,000 unique features and less than 60,000 counts were included in downstream analysis, and nuclei with more than 7.5% mitochondrial reads and 3% ribosomal reads were excluded from analysis. To eliminate batch effects, datasets from unique donors were normalized using the *sctransform* function (version 2) and integrated using the *FindIntegrationAnchors*() in Seurat (version 4.0) in R (**Stuart et al., 2019**). Variables regressed included nFeatures, nCounts, percent mitochondria and ribosomes, and X- and Y-linked genes to avoid sex-associated differences. Dimensional reduction was performed using the *RunPCA*() function to obtain the first 40 principal components, which was determined using *ElbowPlots*() across the first 50 dimensions. To identify clusters, Louvain clustering (Seurat *FindClusters*() function) was performed, and optimal resolution was determined using the *clustree*() function (**Zappia and Oshlack, 2018**) on a range of resolutions between 0.2–1.0, with 0.6 selected for TOs and 0.3 selected for tissue. To identify clusters enriched in combined files of TOs and tissue, dataset integration was performed using Harmony and Louvain clustering (Seurat) performed at a 0.3 resolution, as optimized using *clustree*() (**Zappia and Oshlack, 2018**). Differential expression analysis between clusters was performed using the Wilcoxon rank sum test (Seurat) using *FindAllMarkers*(), with genes with a $\log_2$ fold change threshold set to 0.25 and FDR-adjusted *P*-value <0.05 considered significant. Pseudobulk differential expression analysis was performed using DESeq2 (Love et al,). GO term enrichment was performed with clusterProfiler using *compareCluster*.

## Pseudotime

The *slingshot* package (version 2.6.0) in R was used to determine differentiation trajectories from clusters identified in Seurat with unbiased starting and ending roots (**Street et al., 2018**). The raw counts and above-generated slingshot object were used to run *evaluateK*() with the total number of knots ranging from 3 to 9. The optimal number of knots was determined to be 5. The *fitGAM*() function using *tradeSeq* (1.5.10) was run with this resulting value and gene expression along lineages identified using the *associationTest*() function (**Van den Berge et al., 2020**). Heatmaps of expression changes across lineages were generated using *ComplexHeatmap*() on log transformed counts and rasterized using the *ImageMagick* 'Bessel' filter (**ImageMagick, 2023**). The *plotGenePseudotime*() function was used to visualize raw count gene expression in individual cells across lineages from the *slingshot* object.

## RNA Velocity and Velorama

RNA velocity was performed via the python based program scVelo on snRNAseq data from both placenta tissues of three patients and organoids from three patient-derived TO cell lines (**Bergen et al., 2020**). The organoids were further divided into STB[in] and STB[out] subcategories, resulting in nine total datasets in our analysis. First, we generated UMAPs for each data source (full-term tissue, STB[in], and STB[out]) with corresponding trajectory vectors. For the TOs we included only the CTB, CTB-pf, and STB nucleus types in the analysis. For each data source, the sample datasets were integrated with Scanorama to eliminate dataset-specific batch effects for transcript counts as well as spliced and unspliced transcripts (**Hie et al., 2019a**). The samples were then merged and UMAPs with trajectory vectors were created using the scVelo library. Then, we employed Geosketch to downsample the cell population represented in the UMAP while preserving transcriptomic heterogeneity (**Hie et al., 2019b**). This was done to enhance the clarity and distinction of cells on the UMAP. Second, we inferred gene regulatory networks with Velorama (**Singh et al., 2024**). We compiled lists of human TFs and genes coding for CRs from sources like https://www.factorbook.org/tf/human to use as regulatory genes, along with a selection of highly variable genes and genes of interest from the tissue and organoid datasets to use as our TGs. Velorama was then used to infer the gene regulatory networks under default settings and produced a total of nine regulatory-TG interaction matrices for tissues and organoids raw data sets. Each interaction matrix provides scores that highlight the strength of the relationship between specific regulatory-TG pairs. We then ranked the regulatory-TG pairs by their interaction strength and identified top TFs in each sample, after filtering TFs by number of TGs among the top 500 pairs. Heatmaps were then created based on the overlap of TGs with a score of 1 representing total overlap and a score of 0 indicating no overlap. Finally, top TGs in each cluster were found via sorting by interaction score and plotted as a network analysis with the R package igraph, with the width of the arrow representing the Velorama interaction strength score.

## RNA extraction and bulk RNA-seq analysis

Total RNA from TOs was purified with the Sigma GenElute Universal total RNA purification kit (Sigma-Aldrich, RNB100) following manufacturer's instruction. Purified Total RNA concentration and quality was determined by Thermo scientific Nanodrop one. All RNA samples submitted for bulk RNA-seq were further run for QC evaluation for their RQN (RNA quality number, 10 for all samples) prior to library preparation by the Duke Sequencing and Genomic Technologies (SGT) using KAPA HyperPrep kit (Roche). Sequencing was performed on the NextSeq 1000 XLEAP using P2 flow cell. The reads were aligned to the human reference genome (GRCh38) using QIAGEN CLC Genomics (v20). Differential expression analysis was performed using the DESeq2 package in R with significance cutoff as 0.01 and fold change cutoff at $\log_2 \pm 2$ (*Love et al., 2014*). Files associated with bulk RNA-seq studies have been deposited into the Gene Expression Omnibus (GSE288650). Volcano plots were generated using the EnhancedVolcano package in R (*Kevin Blighe and Lewis, 2024*) or in Graphpad Prism version 9.

## Immunofluorescence in placenta tissue

FFPE tissue sections derived from the same patients sequenced in *Figure 1* were removed from paraffin and rehydrated via an iteration through the following 3 min wash steps (Xylene, Xylene, 1:1 Xylene:100% EtOH, 100% EtOH, 100% EtOH, 95% EtOH, 70% EtOH, 50% EtOH). Slides were then placed under running tap water for 5 min to re-hydrate then transferred to sodium citrate buffer (10 mM sodium citrate, 0.05% Tween 20, pH6) and placed into boiling water for 20 min for antigen revival. Slides were then placed under running tap water for 10 min then transferred into PBS. To perform IF, tissue was permeabilized with 0.5% Triton X-100 in 1 X PBS for 20 min at RT then washed 2 x in 1 X PBS. Reb blood cell autofluorescence was quenched with TrueBlack Lipofuscin Autofluorescence Quencher (Biotium #2300) as per the manufacturer's instructions then washed 2 x in 1 X PBS. Tissue slides were then blocked in blocking buffer (1% BSA in 1 X PBS) for 1 hr at RT. Primary antibodies were then incubated overnight at 4°C in 0.1% BSA in 1 x PBS in the following dilutions: 1:1000 RYBP (Millipore Sigma HPA053357), 1:500 Cytokeratin-7 (Millipore Sigma MABT1490), and 1:200 E-cadherin (BD Biosciences 610181). Slides were rinsed 2 x in 1 X PBS, and incubated in secondary antibodies for 1 hr at 37°C (Invitrogen A-21247, A-11011, A-11001). Slides were rinsed 2 x in 1 X PBS, incubated in 1 µg/ml Hoechst in 1 X PBS for 10 min at RT, and finally washed 1 x in 1 X PBS. PBS was then removed and slides mounted with ProLong Diamond Antifade (Thermo P36965) and dried overnight prior to imaging. Slides were imaged on a Nikon Ti-E stand equipped with a Yokogawa CSU-W1 spinning disk confocal unit with a 40 x Nikon Silicone objective and illuminated with 405/488/565/646 laser sources.

## Cryosectioning and immunofluorescence staining of TOs

TOs were collected in microcentrifuges tubes pre-coated with regular FBS, then rinsed once with 1 X PBS prior to fixing in 4% PFA at RT for 2 hr. Pelleted TOs were washed twice with 1 X PBS and resuspended in 1 X PBS with 0.5 mL 20% (wt/vol) sucrose solution per sample, then transferred to 4°C overnight to allow all the organoid units to pellet into the bottom of sucrose solution. On the following day, 7.5% gelatin (wt/vol)/10% (wt/vol) sucrose embedding solution was pre-warmed at 37°C for 30 min, then organoid units were isolated from sucrose solution into small size molds (7×7 × 5 mm) and polymerize with embedding solution at 4 °C for 20 min before transferring into –80 °C for at least 3 hr prior to cryosectioning.

To section above prepared organoid frozen blocks, the blocks were transferred from –80°C into the cryosection machine (Leica CM1950 Cryostat) at –20°C to equilibrate for 15 min, then 10 µm thickness sections cut. The cryosections were incubated at RT for 15 min before incubation in permeabilization buffer (0.5%<vol/vol >Trition X-100 in 1 X PBS with or without 5%<wt/vol >goat serum) for 45 min at RT. Cryosections were washed and blocked in 5%(v/v) goat serum or BSA/0.1%(v/v) Tween-20 in 1 X PBS for 15 min at RT and then incubated with rabbit anti-human RYBP polyclonal antibody (Sigma, HPA053357), mouse anti-human hCG-β antibody (abcam, ab9582), rat anti-human E-cadherincadherin (Thermo Fisher 14-3249-82) diluted in above-described blocking solution at 4°C overnight. The following day cryosections were washed with 1 X PBS and then incubated for 1 h at RT with Multi-rAb CoraLite Plus 488 Goat anti-mouse (Proteintech, RGAM002), Multi-rAb CoraLite Plus 594 Goat anti-rabbit (Proteintech, RGAR004), and either Donkey Anti-Rat IgG H&L (Alexa Fluor 647) preadsorbed

(Abcam ab150155) recombinant secondary antibodies or Alexa Fluor 647–conjugated phalloidin (Invitrogen, A22287). Cryosections were washed again with 1 X PBS and mounted in Vectashield (Vector Laboratories, H-1200) containing 4',6-diamidino- 2-phenylindole (DAPI). Images were captured using a Olympus Fluoview FV3000 inverted confocal microscope or a Nikon-Yokogawa CSU-W1 spinning disk confocal and contrast-adjusted in Photoshop or Fiji.

## Cryosectioning and RNA fluorescence in situ hybridization (FISH) of TOs

TOs were collected in microcentrifuge tubes pre-coated with 1 mg/mL BSA, rinsed once with 1 X PBS, and fixed in 4% PFA at RT for 30 min. After fixation, TOs were pelleted, washed twice with 1 X PBS, and embedded in OCT within a cryomold. The embedded samples were then flash-frozen in 2-methylbutane and stored at –80°C. Cryosections (10 μm thick) were cut using a cryostat and transferred onto microscope slides (Thermo Fisher 12-550-15). To remove OCT, sections were washed three times with 1 X PBS, followed by permeabilization in 0.5% Triton X-100 in 1 X PBS for 45 min at RT. After two additional PBS washes, sections were incubated for 30 min in 30% formamide wash buffer (30% vol/vol formamide in 2 X saline-sodium citrate (SCC) buffer). For primary probe hybridization, TO sections were incubated overnight at 37°C in 30% hybridization buffer (30% (vol/vol) formamide, 1 mg/mL yeast tRNA, and 10% (wt/vol) dextran sulfate in 2 X SCC buffer) with 1 μM primary FISH probes targeting either PAPPA2 or ADAMTS6 (*Supplementary file 7*). The next day, sections were washed twice with 30% formamide wash buffer at 47°C for 30 min, followed by a 5-min incubation in 10% formamide wash buffer (10% vol/vol formamide in 2 X SCC buffer). Next, TO sections were incubated for 1 hr at 37°C with 5 nM of each secondary probe (bit1_5pCy5, bit11_5pCy5, bit12_5pCy5) in 10% hybridization buffer (*Supplementary file 7*). Following secondary probe hybridization, sections were washed three times in 10% formamide wash buffer for 5 min each at RT. During the second wash, nuclei were counterstained with 1 μg/mL DAPI or Hoechst diluted in 10% hybridization buffer. Finally, TO sections were mounted in Prolong Diamond Antifade (Thermo Fisher P36970) and imaged using a Nikon Ti-E stand equipped with a Yokogawa CSU-W1 spinning disk confocal unit. Imaging was performed with a 100×Nikon silicone objective under 405/646 laser illumination.

## Acknowledgements

We thank Jennifer Gilner, Jillian Hurst, and the Project Hope1000 (Duke University) for providing placental tissue used to derive organoids in this work. We thank Karen Dorman, Neeta Vora, Charles Perou, and Michelle Hayward at UNC Chapel Hill for their assistance in obtaining IRB approval and placenta tissues at UNC CH. This project was supported by NIH AI145828 (CBC), an HHMI Faculty Scholar award (ASG), NSF 743900 (ASG). The organoid work performed by MMK was supported by NIH K00CA245719. We thank the Duke University School of Medicine for the use of the Sequencing and Genomic Technologies Shared Resource and the Translational Genomics Lab at UNC Chapel Hill Lineberger Comprhensive Cancer Center, both of which provided RNA-seq services.

## Additional information

### Funding

| Funder | Grant reference number | Author |
| --- | --- | --- |
| National Institute of Allergy and Infectious Diseases | AI145828 | Carolyn B Coyne |
| National Science Foundation | NSF 743900 | Amy S Gladfelter |
| Howard Hughes Medical Institute | HHMI Faculty Scholar award | Amy S Gladfelter |

The funders had no role in study design, data collection and interpretation, or the decision to submit the work for publication.

## Author contributions
Madeline M Keenen, Conceptualization, Resources, Data curation, Formal analysis, Validation, Investigation, Visualization, Methodology, Writing – original draft, Project administration, Writing – review and editing; Liheng Yang, Data curation, Validation, Investigation, Writing – review and editing; Huan Liang, Rohit Singh, Formal analysis, Investigation, Methodology, Writing – review and editing; Veronica J Farmer, Data curation, Formal analysis, Investigation, Writing – review and editing; Rizban E Worota, Formal analysis, Investigation, Writing – review and editing; Amy S Gladfelter, Conceptualization, Data curation, Formal analysis, Supervision, Funding acquisition, Validation, Investigation, Visualization, Methodology, Writing – original draft, Project administration, Writing – review and editing; Carolyn B Coyne, Conceptualization, Resources, Data curation, Formal analysis, Supervision, Funding acquisition, Validation, Investigation, Visualization, Methodology, Writing – original draft, Project administration, Writing – review and editing

## Author ORCIDs
Madeline M Keenen ⓘ https://orcid.org/0000-0002-9585-3705
Liheng Yang ⓘ https://orcid.org/0000-0001-6842-086X
Huan Liang ⓘ https://orcid.org/0009-0000-4846-2499
Veronica J Farmer ⓘ https://orcid.org/0000-0003-3857-5793
Rizban E Worota ⓘ https://orcid.org/0009-0007-3732-8463
Rohit Singh ⓘ https://orcid.org/0000-0002-4084-7340
Amy S Gladfelter ⓘ https://orcid.org/0000-0002-2490-6945
Carolyn B Coyne ⓘ https://orcid.org/0000-0002-1884-6309

## Ethics
Human subjects: Placenta tissue was collected from patients undergoing scheduled C-sections at UNC Health consented under IRB 21-2055. Inclusion criteria included patients undergoing scheduled C-sections at UNC Health over 18 years of age. Patients were approached in clinic during routine prenatal care. After explaining the study, reviewing the informed consent form, and answering any questions, the patient and consenter signed and dated the consent form. The signed consent form included approval for genomic studies, derivation of cell lines, and publication of results. Patient information, sequencing data, and tissue samples from these experiments was later transferred to Duke under the IRB Pro00113088.

Reviewer #1 (Public review): https://doi.org/10.7554/eLife.101170.3.sa1
Reviewer #1 (Public review): https://doi.org/10.7554/eLife.101170.3.sa2
Reviewer #3 (Public review): https://doi.org/10.7554/eLife.101170.3.sa3
Author response https://doi.org/10.7554/eLife.101170.3.sa4

---

# Additional files

## Supplementary files
Supplementary file 1. Metadata of Tissues and TOs lines used in each experiment.

Supplementary file 2. Gene markers used for cell/nucleus type identification.

Supplementary file 3. Composition of term trophoblast organoid medium (tTOM).

Supplementary file 4. Composition of EVT differentiation medium (EVTM).

Supplementary file 5. sgRNAs sequence used for CRISPR/Cas9 mediated gene editing.

Supplementary file 6. PCR primers used for sequencing validation.

Supplementary file 7. Primary and Secondary FISH probe sequences.

Supplementary file 8. Differentially expressed genes in each STB subtype in the STBin +STBout integrated dataset.

Supplementary file 9. GO terms and representative genes from each STB subtypes in the STBin +STBout integrated dataset.

Supplementary file 10. DEseq analysis of STB subtypes between STBin and STBout in the integrated dataset.

Supplementary file 11. DEseq analysis of bulk sequencing from the WT and knock out TO lines.

Supplementary file 12. Differentially expressed genes in the STB of STBin, STBout, first trimester tissue, and term tissue in the integrated *Figure 6* dataset.

Supplementary file 13. GO terms and representative genes from the STB of STBin, STBout, first trimester tissue, and term tissue in the integrated *Figure 6* dataset.

Supplementary file 14. DEseq analysis of STB between STBin, STBout, first trimester tissue, and term tissue in the integrated *Figure 6* dataset.

MDAR checklist

## Data availability

The datasets analyzed in this paper can be accessed on GEO with the accession number GSE288650. In addition, the processed datasets can be interactively visualized online at https://gladfelterlab.shinyapps.io/PlacentaRNAsequencing/. Code utilized to generate each figure and plasmids utilized to create the RYBP and AFF1 CRISPR KO lines has been uploaded to https://github.com/CoyneLabDuke/snRNASeq-STB-organoid-analysis (copy archived at *CoyneLabDuke, 2025*).

The following dataset was generated:

| Author(s) | Year | Dataset title | Dataset URL | Database and Identifier |
|---|---|---|---|---|
| Keenan MM, Gladfelter AS, Coyne CB | 2025 | Comparative analysis of the syncytiotrophoblast in placenta tissue and trophoblast organoids using snRNA sequencing | https://www.ncbi.nlm.nih.gov/geo/query/acc.cgi?acc=GSE288650 | NCBI Gene Expression Omnibus, GSE288650 |

The following previously published dataset was used:

| Author(s) | Year | Dataset title | Dataset URL | Database and Identifier |
|---|---|---|---|---|
| Liu I, Sun R, Liu F, Li J, Yan L, Zhang J, Xie X, Li D, Wang Y, Li S, Zhu X, Li R, Lu F, Xia Z, Wang H | 2023 | Single-nucleus multi-omic profiling of human placental syncytiotrophoblasts identifies cellular trajectories during pregnancy | https://www.ncbi.nlm.nih.gov/geo/query/acc.cgi?acc=GSE247038 | NCBI Gene Expression Omnibus, GSE247038 |

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
