## [Editor Report · eLife Assessment]

This **important** study uses single-cell transcriptomics to analyze syncytiotrophoblasts in two trophoblast organoid models compared to primary placental tissue, providing **compelling** insights into syncytialization and highlighting the utility of organoid models in placental research. It also serves as an invaluable resource for the field.

---

## [Referee Report · Reviewer #1 (Public review)]

Summary:

This study provides an in-depth analysis of syncytiotrophoblast (STB) gene expression at the single-nucleus (SN) and single-cell (SC) levels, using both primary human placental tissues and two trophoblast organoid (TO) models. The authors compare the older TO model, where STB forms internally (STBin), with a newer model where STB forms externally (STBout). Through a series of comparative analyses, the study highlights the necessity of using both SN and SC techniques to fully understand placental biology. The findings demonstrate that the STBout model shows more differentiated STBs with higher expression of canonical markers and hormones compared to STBin. Additionally, the study identifies both conserved and distinct gene expression profiles between the TO models and human placenta, offering valuable insights for researchers using TOs to study STB and CTB differentiation.

Strengths:

The study offers a comprehensive SC- and SN-based characterization of trophoblast organoid models, providing a thorough validation of these models against human placental tissues. By comparing the older STBin and newer STBout models, the authors effectively demonstrate the improvements in the latter, particularly in the differentiation and gene expression profiles of STBs. This work serves as a critical resource for researchers, offering a clear delineation of the similarities and differences between TO-derived and primary STBs. The use of multiple advanced techniques, such as high-resolution sequencing and trajectory analysis, further enhances the study's contribution to the field.

Weaknesses were addressed during the revision.

The authors effectively addressed my critiques in the rebuttal letter and made corresponding changes in the manuscript. Specifically, they: (1) emphasized the importance of TO orientation in influencing STB nuclear subtype differentiation by adding text to the introduction; (2) clarified the differences in cluster numbers and names between primary tissue and TO data, explaining that each dataset was analyzed independently with separate clustering algorithms and adding clarifying text to the results section; (3) included additional rationale for using SN over SC sequencing, particularly for studying the multinucleated STB; (4) acknowledged that their original evidence was insufficient to definitively determine STBout nuclei differentiation status and removed language suggesting STB-3 as a terminally differentiated subtype, presenting alternative hypotheses in the discussion; and (5) incorporated new figures and clarifications, including RNA-FISH experiments, to validate subtype-specific marker gene expression. Overall, the authors' revisions strengthened the manuscript and aligned well with my critiques.

---

## [Referee Report · Reviewer #1 (Public review)]

Summary:

This study provides an in-depth analysis of syncytiotrophoblast (STB) gene expression at the single-nucleus (SN) and single-cell (SC) levels, using both primary human placental tissues and two trophoblast organoid (TO) models. The authors compare the older TO model, where STB forms internally (STBin), with a newer model where STB forms externally (STBout). Through a series of comparative analyses, the study highlights the necessity of using both SN and SC techniques to fully understand placental biology. The findings demonstrate that the STBout model shows more differentiated STBs with higher expression of canonical markers and hormones compared to STBin. Additionally, the study identifies both conserved and distinct gene expression profiles between the TO models and human placenta, offering valuable insights for researchers using TOs to study STB and CTB differentiation.

Strengths:

The study offers a comprehensive SC- and SN-based characterization of trophoblast organoid models, providing a thorough validation of these models against human placental tissues. By comparing the older STBin and newer STBout models, the authors effectively demonstrate the improvements in the latter, particularly in the differentiation and gene expression profiles of STBs. This work serves as a critical resource for researchers, offering a clear delineation of the similarities and differences between TO-derived and primary STBs. The use of multiple advanced techniques, such as high-resolution sequencing and trajectory analysis, further enhances the study's contribution to the field.

Weaknesses were addressed during the revision.

The authors effectively addressed my critiques in the rebuttal letter and made corresponding changes in the manuscript. Specifically, they: (1) emphasized the importance of TO orientation in influencing STB nuclear subtype differentiation by adding text to the introduction; (2) clarified the differences in cluster numbers and names between primary tissue and TO data, explaining that each dataset was analyzed independently with separate clustering algorithms and adding clarifying text to the results section; (3) included additional rationale for using SN over SC sequencing, particularly for studying the multinucleated STB; (4) acknowledged that their original evidence was insufficient to definitively determine STBout nuclei differentiation status and removed language suggesting STB-3 as a terminally differentiated subtype, presenting alternative hypotheses in the discussion; and (5) incorporated new figures and clarifications, including RNA-FISH experiments, to validate subtype-specific marker gene expression. Overall, the authors' revisions strengthened the manuscript and aligned well with my critiques.

---

## [Referee Report · Reviewer #3 (Public review)]

In this report, Keenen et al. present a thoroughly characterized platform for identifying potential molecular mechanisms regulating syncytiotrophoblast cell functions in placental biology. Application of single cell assessments to identify developmental trajectories of this lineage have been challenging due to the complex, multinucleated structure of the syncytium. The authors provide a comprehensive comparative assessment of term placental tissue and three independent trophoblast organoid models. They use single cell and single nucleus RNA sequencing followed by differential gene expression and pseudotime analyses to identify subpopulations and differentiation trajectories. They further compare the datasets generated in this study to publicly available datasets from first trimester placental tissue. The work is timely as optimization of trophoblast organoids is an evolving topic in placental research. And careful characterization of in vitro models has been noted as essential for model selection and result interpretation in the field.

The study elucidates syncytiotrophoblast nucleus subtypes and proportions in three different organoid models and compares subtypes and gene expression signatures to placental tissues. This work advances the field by demonstrating the utility of different trophoblast organoids to model syncytiotrophoblast differentiation. The in-depth characterization of cell types comprising the different organoid models and how they compare to placental tissue will help to inform model selection for future experimentation in the field. Defining cell composition and cell differentiation trajectories will also aid in data interpretation for data generated by these tissue and model sources. Overall, the conclusions presented in the manuscript are well supported by the data. The figures, as presented, are informative and striking.

The authors present outstanding progress toward their overall aim of identifying, "the underlying control of the syncytiotrophoblast". They identify the chromatin remodeler, RYBP, as well as other regulatory networks that they propose are critical to syncytiotrophoblast development.

The initial study was limited in fully addressing the aim, however, as functional evidence for the contributions of the factors/pathways to syncytiotrophoblast cell development was absent. In a revised version of the manuscript, the authors report the first application of CRISPR-mediated gene silencing in a TO model. They use CRISPR-Cas9-mediated gene targeting to generate RYBP and AFF1 knockout models. Deletion of either RYBP or AFF1 increased STB-2 marker gene expression, as determined using bulk RNA-seq. Future experimentation will assess the distribution of STB nuclear subtypes in the RYBP and AFF1 knockout models and explore the essentiality of RYBP, AFF1, and other identified factors to syncyiotrophoblast development and function.

Localization and validation of the identified factors within tissue and at the protein level will also provide further contextual evidence to address the hypotheses generated. In a revised version of the manuscript, the authors localize STB markers PAPPA2 and ADAMTS6 in TOs using RNA-FISH. Future work will aim to further validate the markers and hypotheses generated from this study.

---

## [Author Response]

The following is the authors’ response to the original reviews

We thank the public reviewers and editors for their insightful comments on the manuscript. We have made the following changes to address their concerns and think the resulting manuscript is stronger as a result. Specifically, we have (1) added RNA FISH data of specific STB-2 and STB-3 RNA markers to confirm their distribution changes between STB^in^ and STB^out^ TOs, (2) removed language throughout the text that refer to STB-3 as a terminally differentiated nuclear subtype, and (3) generated CRISPR-mediated knock-outs of two genes identified by network analysis and validated their rolse in mediating STB nuclear subtype gene expression.

**Reviewer #1 (Public review):**
Strengths:The study offers a comprehensive SC- and SN-based characterization of trophoblast organoid models, providing a thorough validation of these models against human placental tissues. By comparing the older STB^in^ and newer STB^out^ models, the authors effectively demonstrate the improvements in the latter, particularly in the differentiation and gene expression profiles of STBs. This work serves as a critical resource for researchers, offering a clear delineation of the similarities and differences between TO-derived and primary STBs. The use of multiple advanced techniques, such as high-resolution sequencing and trajectory analysis, further enhances the study's contribution to the field.

Thank you for your thoughtful review—we appreciate your recognition of our efforts to comprehensively validate trophoblast organoid models and highlight key advancements in STB differentiation and gene expression.

Weaknesses:While the study is robust, some areas could benefit from further clarification.(1) The importance of the TO model's orientation and its impact on outcomes could be emphasized more in the introduction.

We agree that TO orientation may significantly influence STB nuclear subtype differentiation. As the STB is critical for both barrier formation and molecular transport in vivo, lack of exposure to the surrounding media in STB^in^ TOs in vitro could compromise these functions and the associated environmental cues that influence STB nuclear differentiation. We have added text to the introduction to highlight this point (lines 117-120).

(2) The differences in cluster numbers/names between primary tissue and TO data need a clearer explanation, and consistent annotation could aid in comparison.

Thank you for highlighting that the comparisions and cluster annotations need clarification. In Figure 1, we did not aim to directly compare CTB and STB nuclear subtypes between TOs and tissue. Each dataset was analyzed independently, with clusters determined separately and with different resolutions decided via a clustering algorithm (Zappia and Oshlack, 2018). For example, for the STB, this approach identified seven subtypes in tissue but only two in TOs, making direct comparison challenging. To address this challenge, we integrated the SN datasets from TOs and tissue in Figure 6. This integration allowed us to directly compare gene expression between the sample types and examine the proportions within each STB subtype. Similarly, in Figure 2, direct comparison of individual CTB or STB clusters across the separate datasets is challenging (Figures 2A-C) due to differences in clustering. To overcome this, we integrated the datasets to compare cluster gene expression and relative proportions (Figures 2D-E). Nonetheless, to address the reviewers concern we have added text to the results section to clarify that subclusters of CTB and STB between datasets should not be directly compared until the datasets are integrated in Figure 2D-E and Figure 6 (lines 166-167).

(3) The rationale for using SN sequencing over SC sequencing for TO evaluations should be clarified, especially regarding the potential underrepresentation of certain trophoblast subsets.

This is an important point as the challenges of studying a giant syncytial cell are often underappreciated by researchers that study mononucleated cells. We have added text to the introduction to clarify why traditional single cell RNA sequencing techniques were inadequate to collect and characterize the STB (lines 91-93).

(4) Additionally, more evidence could be provided to support the claims about STB differentiation in the STB^out^ model and to determine whether its differentiation trajectory is unique or simply more advanced than in STB^in^.

Our original conclusion that STB^out^ nuclei are more terminally differentiated than STB^in^ was based on two observations: (1) STB^out^ TOs exhibit increased expression of STB-specific pregnancy hormones and many classic STB marker genes and (2) STB^out^ nuclei show an enrichment of the STB-3 nuclear subtype, which appears at the end of the slingshot pseudotime trajectory. However, upon consideration of the reviewer comments, we agree that this evidence is not sufficient to definitively distinguish if STB^out^ nuclei are more advanced or follow a unique differentiation trajectory dependent on new environmental cues. Pseudotime analyses provided only a predictive framework for lineage tracing, and these predictions must be experimentally validated. Real-time tracking of STB nuclear subtypes in TOs would require a suite of genetic tools beyond the scope of this study. Therefore, to address the reviewers' concerns we have removed language suggesting that STB-3 is a terminally differentiated subtype or that STB^out^ nuclei are more differentiated than STB^in^ nuclei throughout the text until the discussion. Therein we present both our original hypothesis (that STB nuclei are further differentiated in STB^out^) and alternative explanations like changing trajectories due to local environmental cues (lines 619-625).

**Reviewer #2 (Public review):**
Strengths:(1) The use of SN and SC RNA sequencing provides a detailed analysis of STB formation and differentiation.(2) The identification of distinct STB subtypes and novel gene markers such as RYBP offers new insights into STB development.

Thank you for highlighting these strengths—we appreciate your recognition of our use of SN and SC RNA sequencing to analyze STB differentiation and the discovery of distinct STB subtypes and novel gene markers like RYBP.

Weaknesses:(1) Inconsistencies in data presentation.

We address the individual comments of reviewer 2 later in this response.

(2) Questionable interpretation of lncRNA signals: The use of long non-coding RNA (lncRNA) signals as cell type-specific markers may represent sequencing noise rather than true markers.

We appreciate the reviewer’s attention to detail in noticing the lncRNA signature seen in many STB nuclear subtypes. However, we disagree that these molecules simply represent sequencing noise. In fact, may studies have rigorously demonstrated that lncRNAs have both cell and tissue specific gene expression (e.g., Zhao et al 2022, Isakova et al 2021, Zheng et al 2020). Further, they have been shown to be useful markers of unique cell types during development (e.g., Morales-Vicente et al 2022, Zhou et al 2019, Kim et al 2015) and can enhance clustering interpretability in breast cancer (Malagoli et al 2024). Many lncRNAs have also been demonstrated to play a functional role in the human placenta, including H19, MEG3, and MEG8 (Adu-Gyamfi et al 2023) and differences are even seen in nuclear subtypes in trophoblast stem cells (Khan et al 2021). Therefore, we prefer to keep these lncRNA signatures included and let future researchers test their functional role.

To improve the study's validity and significance, it is crucial to address the inconsistencies and to provide additional evidence for the claims. Supplementing with immunofluorescence staining for validating the distribution of STB_in, STB_out, and EVT_enrich in the organoid models is recommended to strengthen the results and conclusions.

Each general trophoblast cell type (CTB, STB, EVT) has been visualized by immunofluorescence by the Coyne laboratory in their initial papers characterizing the STB^in^, STB^out^, and EVT^enrich^ models (Yang et al, 2022 and 2023). We agree that it is important to validate the STB nuclear subtypes found in our genomic study. However, one challenge in studying a syncytia is that immunofluorescence may not be a definitive method when the nuclei share a common cytoplasm. This is because protein products from mRNAs transcribed in one nucleus are translated in the cytoplasm and could diffuse beyond sites of transcription. Therefore, RNA fluorescence in situ hybridization (RNA-FISH) is instead needed. While a systematic characterization of the spatial distribution of the many marker genes found each subtype is outside the scope of this study, we include RNA-FISH of one STB-2 marker (PAPPA2) and one STB-3 marker (ADAMTS6) in Figure 3F-G and Supplemental Figure 3.3. This demonstrates there is an increase in STB-2 marker gene expression in STB^in^ TOs and an increase in STB-3 marker gene expression in STB^out^ TOs.

**Reviewer #3 (Public review):**
The authors present outstanding progress toward their aim of identifying, "the underlying control of the syncytiotrophoblast". They identify the chromatin remodeler, RYBP, as well as other regulatory networks that they propose are critical to syncytiotrophoblast development. This study is limited in fully addressing the aim, however, as functional evidence for the contributions of the factors/pathways to syncytiotrophoblast cell development is needed. Future experimentation testing the hypotheses generated by this work will define the essentiality of the identified factors to syncytiotrophoblast development and function.

We thank the reviewer for their thoughtful assessment, constructive feedback, and encouraging comments. We acknowledge that the initial manuscript primarily presented analyses suggesting correlations between RYBP and other factors identified in the gene network analysis and STB function. Understanding how gene networks in the STB are formed and regulated is a long-term goal that will require many experiments with collaborative efforts across multiple research groups.

Nonetheless, to address this concern we have knocked out two key genes, RYBP and AFF1, in TOs using CRISPR-Cas9-mediated gene targeting. Bulk RNA sequencing of STB^in^ TOs from both wild-type (WT) and knockout strains revealed that deletion of either gene caused a statistically significant decrease in the expression of the pregnancy hormone *human placental lactogen* and an increase in the expression of several genes characteristic of the oxygen-sensing STB-2 subtype, including *FLT-1*, *PAPPA2*, *SPON2*, and *SFXN3*. These findings demonstrate that knocking out RYBP or AFF1 results in an increase in STB-2 marker gene expression and therefore play a role in inhibiting their expression in WT TOs (Figure 5D-E and supplemental Figure 5.2). We also note that this is the first application of CRISPR-mediated gene silencing in a TO model.

Future work will visualize the distribution of STB nuclear subtypes in these mutants and explore the mechanistic role of RYBP and AFF1 in STB nuclear subtype formation and maintenance. However, these investigations fall outside the scope of the current study.

Localization and validation of the identified factors within tissue and at the protein level will also provide further contextual evidence to address the hypotheses generated.

We agree that visualizing STB nuclear subtype distribution is essential for testing the many hypotheses generated by our analysis. To address this, we have included RNA-FISH experiments for two STB subtype markers (*PAPPA2* for STB-2 and *ADAMTS6* for STB-3) in TOs. These experiments reveal an increase in *PAPPA2* expression in STB^in^ TOs and an increase in *ADAMTS6* expression in STB^out^ TOs (Figure 3F-G and Supplemental Figure 3.3). Genomic studies serve as powerful hypothesis generators, and we look forward to future work—both our own and that of other researchers—to validate the markers and hypotheses presented from our analysis.

**Recommendations for the authors:**

**Reviewing Editor Comments:**
We strongly encourage the authors to further strengthen the study by addressing all reviewers' comments and recommendations, with particular attention to the following key aspects:(1) Clarifying the uniqueness of the STB differentiation trajectory between STB^in^ and STB^out^, and determining whether STB^out^ represents a more advanced stage of differentiation compared to STB^in^. It is also important to specify which developmental stage of placental villi the STB^out^ and STB^in^ are simulating.

We have revised the manuscript to remove definitive language claiming that STB-3 represents a terminally differentiated subtype or that STB^out^ nuclei are more differentiated than STB^in^ nuclei. Instead, we now present our hypothesis and alternative explanations in the discussion (lines 619-625), and emphasize the need for experimental validation of pseudotime predictions to test these hypotheses.

(2) Utilizing immunofluorescence to validate the distribution of cell types in the organoid models.

The Coyne lab has previously performed immunofluorescence of CTB and STB markers in STB^in^ and STB^out^ TOs (Yang et al 2023). The syncytial nature of STBs complicates immunofluorescence-based validation of the STB nuclear subtypes due translating proteins all sharing a single common cytoplasm and therefore being able to diffuse and mix. Instead, we performed RNA-FISH for two STB subtype markers (PAPPA2, STB-2 and ADAMTS6, STB-3), which showed subtype-specific nuclear enrichment in STB^in^ and STB^out^ TOs, respectively (Figure 3F-G and Supplemental Figure 3.3).

(3) Addressing concerns regarding the use of lncRNA as cell marker genes. Employing canonical markers alongside critical TFs involved in differentiation pathways to perform a more robust cell-type analysis and validation is recommended.

As discussed in detail above, we maintain that lncRNAs are valuable markers, supported by their demonstrated roles in cell and tissue specificity and placental function. These signatures provide important insights and hypotheses for future research, and we have clarified this rationale in the revised manuscript.

**Reviewer #1 (Recommendations for the authors):**
(1) The authors have presented an extensive SC- and SN-based characterization of their improved trophoblast TO model, including a comparison to human placental tissues and the previous TO iteration. In this way, the authors' work represents an invaluable resource for investigators by providing thorough validation of the TO model and a clear description of the similarities and differences between primary and TO-derived STBs. I would suggest that the authors reshape the study to further highlight and emphasize this aspect of the study.

We thank the reviewer for their thoughtful recommendation and agree that our datasets will serve as an invaluable resource for comparing in vitro models to in vivo gene expression. However, extensive validation is required to make definitive conclusions about the extent to which these systems mirror one another and where they diverge. For this reason, in this manuscript, we have focused on characterizing STB subtypes to provide a foundational understanding of the model and this poorly characterized subtype.

(2) Introduction, Paragraph 3: What is the importance of orientation for the trophoblast TO model? The authors may consider removing some of the less important methodologic details from this paragraph and including more emphasis on why their TO model is an improvement.

Text has been added to this paragraph to highlight the importance of outward facing STB orientation, which is essential to mirror the STB’s transport function in vitro (lines 118-120).

(3) Results, Figure 1: In addition to the primary placental tissue plots showing all cell populations, it may be useful to have side-by-side versions of similar plots showing only the trophoblast subsets, so that the primary and TO data could be more easily compared visually.

This has been implemented and added to the Supplemental Figure 1.4.

(4) Results, Figure 1: In simple terms, what is the reason for ending up with different cluster numbers/names from the primary tissue and TO? Would it be possible to apply the same annotation to each (at least for trophoblast types) and thus allow direct comparison between the two?

As described above, each dataset was separately analyzed and clusters determined with an algorithm to determine the optimal clustering resolution. Therefore, the number of clusters between each dataset cannot be directly compared until the SN TO and tissue datasets are integrated together in Figure 6. We have added text to the manuscript to make it clear that they should not be compared except for in bulk number until this point (230-232).

(5) Results, Figure 2: For subsequent evaluation of different in vitro TO conditions, did the authors use only SN sequencing because they wanted to focus on STB? Based on Figure 1, it seems some CTB subsets would be underrepresented if using only SN. Given that the authors look at both STB and CTB in their different TOs, is this an issue?

The CTB clusters that showed the greatest divergence between SC and SN datasets were those associated with mitosis and the cell cycle, likely due to nuclear envelope breakdown interfering with capture by the 10x microfluidics pipeline. While cytoplasmic gene expression provides valuable insights into CTB function, our manuscript focuses on the STB starting from Figure 2. Since the STB is captured exclusively by the SN dataset, we concentrated on this approach to streamline our analysis.

(6) Results, Figure 3: What do the authors consider to be the primary contributing factors for why the STB subsets display differential gene expression between STB^in^ and STB^out^? Is this due primarily to the cultural conditions and/or a result of the differing spatial arrangement with CTBs?

This is an intriguing question that is challenging to disentangle because the culture conditions are integral to flipping the orientation. The two primary factors that differ between STB^in^ and STB^out^ TOs are the presence of extracellular matrix in STB^in^ and direct exposure to the surrounding media in STB^out^. We believe these environmental cues play a significant role in shaping the gene expression of STB subsets. Fully disentangling this relationship would require a method to alter the TO orientation without changing the culture conditions. While this is an exciting direction for future research, it falls outside the scope of the present study.

(7) Results, Figure 4: The authors' analysis indicates that the STB nuclei from the STB^out^ TO are likely "more differentiated" than those in STB^in^ TO. Could the authors provide some qualitative or quantitative support for this? Is the STB^out^ differentiated phenotype closer to what would be observed in a fully formed placenta?

As discussed earlier, we agree with the reviewers that this claim should be removed from the text outside of the discussion.

(8) Results, Figure 5: Based on the trajectory analysis, do the authors consider that the STB from STB^out^ TO are simply further along the differentiation pathway compared to those from STB^in^ TO, or do the STB from STB^out^ TO follow a differentiation pathway that is intrinsically distinct from STB^in^ TO?

We think the idea of an intrinsically distinct pathway is a fascinating alternative hypothesis and have added it into the discussion. We do not find the pseudotime currently allows us to answer this question without additional experiments, so we have removed claims that the STB^out^ STB nuclei are further along the differentiation pathway.

(9) Results, Figure 6: A notable difference between the STB^out^ TO and the term tissue is that the CTB subsets are much more prevalent. Is this simply a scale difference, i.e. due to the size of the human placenta compared to the limited STB nuclei available in the STB^out^ TO? Or are there other contributing factors?

The proportion of CTB to STB nuclei in our term tissue (9:1) aligns with expectations based on stereological estimates. We believe the relatively low number of CTB nuclei in our dataset is due to the need for a larger sample size to capture more of this less abundant cell type. Since the primary focus of this paper is on STB, and we analyzed over 4,000 STB nuclei, we do not view this as a limitation. However, future studies utilizing SN to investigate term tissue should account for the abundance of STB nuclei and plan their sampling carefully to ensure sufficient representation of CTB nuclei if this is a desired focus.

**Reviewer #2 (Recommendations for the authors):**
(1) The color annotations for cell types in Figure 2 are inconsistent between the different panels, and the term "Prolif" in Figure 2E is not explained by the authors.

We chose colors to enhance visibility on the UMAP. We do not wish readers to make direct comparisons between the different CTB or STB subtypes of the sample types until the datasets are integrated in Figure 2D. This is because an algorithm for the clustering resolution has been chosen independently for each dataset. Cluster proportions are better compared in the integrated datasets in Figure 2D. We have added text to the results section to make this clear to the reader (lines 166-167).

(2) In Figure 3 and Supplementary Figures 1.3, the authors frequently present long non-coding RNA (lncRNA) signals as cell type-specific markers in the bubble plots. These signals are likely sequencing noise and may not accurately represent true markers for those cell types. It is recommended to revise this interpretation.

As referenced above, there are many examples of lncRNAs that have biological and pathological significance in the placenta (H19, Meg3, Meg8) and lncRNAs often have cell type specific expression that can enhance clustering. We prefer to keep these signatures included and let future researchers determine their biological significance.

(3) In Figure 3C, the authors performed pathway enrichment analysis on the STB subtypes after integrating STB_in and STB_out organoids. The enrichment of the "transport across the blood-brain barrier" pathway in the STB-3 subtype does not align with the current understanding of STB cell function. Please provide corresponding supporting evidence. Additionally, please verify whether the other functional pathways represent functions specific to the STB subtypes.

Interestingly, many of the genes categorized under “transport across the blood-brain barrier” are transporters shared with “vascular transport.” These include genes involved in the transport of amino acids (SLC7A1, SLC38A1, SLC38A3, SLC7A8), molecules essential for lipid metabolism (SLC27A4, SLC44A1), and small molecule exchange (SLC4A4, SLC5A6). Given that the vasculature, the STB, and the blood-brain barrier all perform critical barrier functions, it is unsurprising that molecules associated with these GO terms are enriched in the STB-3 subtype, which expresses numerous transporter proteins. Since the transport of materials across the STB is a well-established function, we have not included additional supporting evidence but have clarified the genes associated with this GO term in the text (lines 392-394 and supplemental Table 9).

(4) The pseudotime heatmap in Figure 4B is not properly arranged and is inconsistent with the differentiation relationships shown in Figure 4A. It is recommended to revise this.

We are uncertain which aspect of the heatmap in Figure 4A is perceived as inconsistent with Figure 4B. One distinction is that pseudotime in Figure 4A is normalized from 0 to 100 to fit the blue-to-yellow-to-red color scale, whereas in Figure 4B, the color scale is not normalized and the color bar ranging from white to red. This difference reflects our intent to simplify Figure 4B-C, as the abundance of color between cell types and gene expression changes required a streamlined representation to ensure the figure remained clear and easy to interpret. This is classically done in the field and consistent with the default code in the slingshot package.

(5) In Figures 4C and 4D, although RYBP is highly expressed in STB, it is difficult to support the conclusion that RYBP shows the most significant expression changes. It is recommended to provide additional evidence.

The claim that RYBP exhibits the most significant expression changes was based on p-value ordering of genes associated with pseudotime via the associationTest function in slingshot and not with immunofluorescence data. The text has been revised to make this distinction clear (lines 390-393).

(6) In Figure 4E, staining for CTB marker genes is missing, and in Figure 4F, CYTO is difficult to use as a classical STB marker. It is recommended to use the CGBs antibody from Figure 4E as a STB marker for staining to provide evidence.

We have revised the Figure 5B-C to use e-Cadherin as a CTB marker gene in TOs and CGB antibody as a marker of STB.

In tissue, however, obtaining a good STB marker that does not overlap with the RYBP antibody (rabbit) in term tissue is difficult as the STB downregulates hCG expression closer to term to initiate contractions. SDC1 is often used but only labels the plasma membrane so does not help in distinguishing the STB cytoplasm. We have added an image of cytokeratin, e-Cadherin, and the STB marker ENDOU to validate that our current approach with e-Cadherin and cytokeratin allows us to accurately distinguish between CTB and STB cells.

(7) The velocity results in Figure 5A do not align with the differentiation relationships between cells and contradict the pseudotime results presented in Figure 4 by the authors.

The reviewer raises an interesting observation regarding the velocity map in Figure 5A, which appears to show a bifurcation into two STB subtypes. This observation aligns with similar findings reported in tissue by our colleagues (Wang et al., 2024). However, given the low number of CTB cells in our tissue dataset, we were cautious about making definitive conclusions about pseudotime without a larger sample size. Notably, the RNA velocity map closely resembles the pseudotime trajectory in TOs, with CTB transitioning into the CTB-pf subtype and subsequently into the STB. One potential explanation for discrepancies between tissue and TOs is the difference in nuclear age: nuclei in tissue can be up to nine months old, whereas those in TOs are only hours or days old. It is possible that the lineage in TOs could bifurcate if cultured for longer than 48 hours, but our current dataset captures only the early stages of the STB differentiation process. While exploring these hypotheses is fascinating, they are beyond the scope of this current study.

**Reviewer #3 (Recommendations for the authors):**
Amazing work - I greatly enjoyed reading the manuscript. Here are a few questions and suggestions for consideration:Evidence presented throughout the results sections hints that the organoids may represent an earlier stage of placental development compared to the term. Increased hCG gene expression is observed, but as noted expression is decreased in term STB. STB:CTB ratios are also higher at term compared to the first trimester, etc. It was difficult to conclude definitively based on how data is presented in Fig 6 and discussed. Maybe there is no clear answer. Perhaps the altered cell type ratios in the organoid models (e.g., few STB in EVT enrich conditions) impact recapitulation of the in vivo local microenvironment signaling. As such, can the authors speculate on whether cell ratios could be strategically leveraged to model different gestational time points?Along these same lines, syncytiotrophoblast in early implantation (before proper villi development) is often described as invasive and later at the tertiary villi stage defined by hormone production, barrier function, and nutrient/gas exchange. Do the authors think the different STB subtypes captured in the organoid models represent different stages/functions of syncytiotrophoblast in placental development?Minor Comments(1) Please clarify what the third number represents in the STB:CTB ratio (e.g., 1:3:1 and 2:5:1). EVT?

The first number is a decimal point and not a colon (ie 1.3 and 2.5). Therefore these numbers are to be read as the STB:CTB ratio is 1.3 to 1 or 2.5 to 1.

(2) Could consider co-localizing RYBP in term tissue with a syncytio-specific marker like CGB used for organoids (Fig 4F).

We addressed this concern in comment 6 to reviewer 2.

(3) Recommend defining colors-which colors represent which module in Figure 5C in the legend and main body text. I see the labels surrounding the heatmap in 5B, but defining colors in text (e.g. cyan, magenta, etc.) would be helpful. Do the gray circles represent targets that don't belong to a specific module? Are the bolded factor names based on a certain statistical cutoff/defining criteria or were they manually selected?

The text of both the results and figure legends has been revised to clarify these points.

(4) Data Availability: It would be helpful to provide supplemental table files for analyses (e.g., 5C to list the overlapping relationships in TGs for each TF/CR (5C) and 3E/6F to list DEG genes in comparisons).

Supplemental files for each analysis have been added (Supplemental Table 8-14). In addition, the raw and processed data is available on GEO and we have created an interactive Shiny App so people without coding experience can interact with each dataset (lines 917-919).

(5) “...and found that each sample expressed these markers (Figure 6D), suggesting..." Consider clarifying "these".

Text has been added to refer to a few of these marker genes within the text (line 540).

Citations

(1) Zappia L, Oshlack A. Clustering trees: a visualization for evaluating clusterings at multiple resolutions. GigaScience. 2018;7(7):giy083. PMCID: PMC6057528

(2) Zhou J, Xu J, Zhang L, Liu S, Ma Y, Wen X, Hao J, Li Z, Ni Y, Li X, Zhou F, Li Q, Wang F, Wang X, Si Y, Zhang P, Liu C, Bartolomei M, Tang F, Liu B, Yu J, Lan Y. Combined Single-Cell Profiling of lncRNAs and Functional Screening Reveals that H19 Is Pivotal for Embryonic Hematopoietic Stem Cell Development. Cell Stem Cell. 2019;24(2):285-298.e5. PMID: 30639035

(3) Malagoli G, Valle F, Barillot E, Caselle M, Martignetti L. Identification of Interpretable Clusters and Associated Signatures in Breast Cancer Single-Cell Data: A Topic Modeling Approach. Cancers. 2024;16(7):1350. PMCID: PMC11011054

(4) Adu-Gyamfi EA, Cheeran EA, Salamah J, Enabulele DB, Tahir A, Lee BK. Long non-coding RNAs: a summary of their roles in placenta development and pathology†. Biol Reprod. 2023;110(3):431–449. PMID: 38134961

(5) Zheng M, Hu Y, Gou R, Nie X, Li X, Liu J, Lin B. Identification three LncRNA prognostic signature of ovarian cancer based on genome-wide copy number variation. Biomed Pharmacother. 2020;124:109810. PMID: 32000042

(6) Khan T, Seetharam AS, Zhou J, Bivens NJ, Schust DJ, Ezashi T, Tuteja G, Roberts RM. Single Nucleus RNA Sequence (snRNAseq) Analysis of the Spectrum of Trophoblast Lineages Generated From Human Pluripotent Stem Cells in vitro. Front Cell Dev Biol. 2021;9:695248. PMCID: PMC8334858

(7) Isakova A, Neff N, Quake SR. Single-cell quantification of a broad RNA spectrum reveals unique noncoding patterns associated with cell types and states. Proc Natl Acad Sci United States Am. 2021;118(51):e2113568118. PMCID: PMC8713755

(8) Morales-Vicente DA, Zhao L, Silveira GO, Tahira AC, Amaral MS, Collins JJ, Verjovski-Almeida S. Singlecell RNA-seq analyses show that long non-coding RNAs are conspicuously expressed in Schistosoma mansoni gamete and tegument progenitor cell populations. Front Genet. 2022;13:924877. PMCID: PMC9531161

(9) Kim DH, Marinov GK, Pepke S, Singer ZS, He P, Williams B, Schroth GP, Elowitz MB, Wold BJ. Single-Cell

Transcriptome Analysis Reveals Dynamic Changes in lncRNA Expression during Reprogramming. Cell Stem Cell. 2015;16(1):88–101. PMCID: PMC4291542

(10) Yang L, Liang P, Yang H, Coyne CB. Trophoblast organoids with physiological polarity model placental structure and function. bioRxiv. 2023;2023.01.12.523752. PMCID: PMC9882188